# GREATS: Online Selection of High-Quality Data for LLM Training in *Every* Iteration

**Jiachen T. Wang**[*]
Princeton University
tianhaowang@princeton.edu

**Tong Wu**
Princeton University
tongwu@princeton.edu

**Dawn Song**
UC Berkeley
dawnsong@cs.berkeley.edu

**Prateek Mittal**
Princeton University
pmittal@princeton.edu

**Ruoxi Jia**[*]
Virginia Tech
ruoxijia@vt.edu

## Abstract

Online batch selection methods offer an adaptive alternative to static training data selection by dynamically selecting data batches during training. However, existing methods either rely on impractical reference models or simple heuristics that may not capture true data informativeness. To address these limitations, we propose *GREedy Approximation Taylor Selection* (GREATS), a principled and efficient online batch selection method that applies greedy algorithm to optimize the data batch quality approximated by Taylor expansion. We develop a series of techniques to scale GREATS to large-scale model training. Extensive experiments with large language models (LLMs) demonstrate that GREATS significantly improves training convergence speed and generalization performance. Our codebase is publically available at https://github.com/Jiachen-T-Wang/GREATS.

## 1 Introduction

Large language models (LLMs) are of paramount importance in today's technological landscape. However, the extensive training times, often spanning weeks or even months, pose challenges such as prolonged development cycles and increased resource consumption. Moreover, these models are trained on massive data collected from the open world, which can include low-quality, redundant, and biased information. This underscores the need for effective selection of high-value training data.

**Online batch selection: an adaptive variant of data selection at the batch level.** Online batch selection methods aim to improve data selection by dynamically choosing data during the training process. At each training iteration, these methods leverage the partially trained model to determine which data to select for the current training iteration from a sampled batch, thereby adapting to the model's learning progress and focusing on the most informative examples *for the model's current state*. In contrast to static data selection methods (e.g., [11, 43, 42, 45]), which select training data only once prior to the training process, online batch selection allows for a more adaptive and dynamic approach to data selection. By continuously updating the selection criteria based on the model's progress, online batch selection can identify the most relevant and informative examples at each stage of training, potentially leading to faster convergence and better generalization performance. Moreover, online batch selection operates on smaller batches of data, reducing the need for cumbersome data preprocessing and enabling more efficient use of computational resources compared to static data selection methods that process the entire dataset upfront.

---

[*]Correspondence to **Jiachen T. Wang** and **Ruoxi Jia**.

38th Conference on Neural Information Processing Systems (NeurIPS 2024).

However, existing online batch selection algorithms exhibit significant limitations and have found minimal success with LLMs. Reference-model-based batch selection methods [30, 9] rely on additional reference models. In [30], the reference models are trained on a substantial amount of hold-out data. This leads to considerable computational costs and reduces the amount of data available for training the main model. [9] utilize publicly available large-scale pre-trained models that already achieve very high performance on the targeted downstream tasks as reference models. Even if this assumption holds true in practice, the algorithm requires a Bayesian treatment and querying the reference models for every sample in the candidate batch at each iteration. These operations are computationally expensive, making the approach impractical for large-scale LLM training. Reference-model-free methods prioritize challenging samples based on metrics like high loss [28, 20] or large gradient norm [23]. While some of these methods are computationally efficient and practical to implement, they often rely on simple heuristics that may not capture the true informativeness or relevance of the examples. As a result, these methods often fall short in terms of performance and may even underperform compared to simple uniform selection in some cases. This highlights the need for more efficient and principled online batch selection techniques that can identify the most informative examples based on a deeper understanding of the model's learning dynamics and the relationships between the examples.

In this work, we propose *GREedy Approximation Taylor Selection* (GREATS), which addresses the limitations of existing methods and significantly improves the convergence speed and generalization performance of language model training. We summarize our contributions as follows.

**I. Principled Formulation for Optimal Batch Selection.** We introduce a principled formulation of the online batch selection problem as a set utility function optimization task. Given a small set of validation data from the target domain, the utility function measures the reduction in loss achieved by updating the model with a selected subset of the training batch. Unlike previous methods that rely on heuristics, this framework aims to directly optimize the model's performance on the validation set, ensuring the selection of informative and diverse examples.

**II. Efficient Approximations for Scalable Batch Selection.** The set function optimization formulation naturally leads to a greedy algorithm that iteratively selects the most informative examples based on their marginal contribution to the model's performance. However, directly applying the greedy algorithm to optimize the validation performance involves updating model with each potential candidate training point and checking the validation performance, which is computationally inefficient. To tackle this challenge, we propose to leverage Taylor expansions to approximate the variation of validation loss in one-step gradient descent. The key insight is that the impact of a training example on the model's validation loss can be efficiently approximated using gradient inner-products between the training examples and the validation data. This approximation eliminates the need for expensive model updates and validation loss evaluations for each candidate subset.

**III. Online Batch Selection at the Speed of Regular Training.** A direct implementation of GREATS would require computing per-sample gradients, which is computationally expensive. To address this challenge, we develop a novel technique called *"ghost inner-product"* that allows for the efficient computation of pairwise gradient inner-products without the need to instantiate any model-sized vectors. As gradient inner-products arise in various machine learning algorithms and applications beyond data selection, this technique may be of independent interest.

**IV. Comprehensive Evaluations.** We conduct extensive experiments on various language modeling tasks to thoroughly assess the performance of GREATS. We show that GREATS consistently speeds up training convergence and improves generalization performance across different models, training datasets, and evaluation datasets, even with a limited number of validation points. Furthermore, we show that GREATS can provide benefits even in the pretraining setting, where the validation data comes from the same domain as the training dataset. This highlights the robustness and versatility of our approach in various learning scenarios. In addition to its performance benefits, we empirically confirm that GREATS, equipped with the "ghost inner-product" technique, achieves a runtime comparable to regular training. This underscores the practical feasibility of our approach.

## 2  Related Works

**Online Batch Selection.** Few studies have investigated the use of online batch selection to enhance the training of models before the era of large language models. [28, 23, 20] examined selecting the

"hard examples" based on their gradient norm or maximum sample loss. While some of these methods are computationally efficient, they often depend on simple heuristics that cannot represent the true informativeness. [30, 9] suggested using additional reference models to more accurately estimate the importance of samples. However, recently [22] demonstrated these methods are computationally expensive and cannot directly apply to large language models.

**Static Data Selection for Large-scale Models.** Recently, there has been a growing interest in design methods to select data *before* training foundation models. We point the readers to [2] for a comprehensive literature review. These works select training data only *once*, prior to the training process. This is primarily motivated by efficiency concerns, as the time spent on data selection can be amortized over a large number of training steps. However, the non-adaptive nature of this single-step selection often results in suboptimal performance, as the selected data may not be the most informative or diverse throughout the entire training process [40]. Moreover, these algorithms often require extensive and complex data preprocessing steps. Some of the data selection algorithms even require training an additional model solely for the purpose of data selection [43, 44], which introduces additional computational costs and implementation complexity to the training pipeline. These drawbacks emphasize the need for more efficient and *adaptive* data selection techniques that can dynamically identify the most informative and relevant examples throughout the training process.

**Online Domain Reweighting.** Recent work has explored online methods for dynamically re-weighting domains during language model pre-training. Compared with online batch selection, this approach operates at a coarser granularity by focusing on data source-level selection rather than individual examples, and typically updates domain weights less frequently. Similar to this work, [12] uses a gradient-based influence estimation to update domain weights. [3] uses training loss as a reward signal to adapt domain sampling probabilities. This has been recently refined by [21] through a scaling law-based method.

## 3 Background

In this section, we introduce the setup of online batch selection and the concept of a utility function. We then discuss the limitations of existing scoring and top-$k$ paradigm in the literature.

**Set-up of online batch selection.** Given a training dataset $\mathcal{D}_{\text{tr}} = \{z_i\}_{i=1}^N$, a deep learning model is usually being trained to minimize the training loss $\sum_{i=1}^N \ell(w, z_i)$ via an iterative optimization procedure such as stochastic gradient descent (SGD). Starting with an initial model $w_0$, during an iteration $t$, a batch $S$ of the training points is being used, and update the model parameters from $w_t$ to $w_{t+1}$ via $w_{t+1} := w_t - \eta_t \sum_{z \in S} \nabla \ell(w_t, z)$ where $\eta_t$ is the learning rate at iteration $t$.[1] A complete run of neural network training thus consists of model checkpoints $\{w_0, w_1, \ldots, w_T\}$. In the setting of online batch selection, a large batch $\mathcal{B}_t = \{z_1, \ldots, z_B\}$ is being sampled from the training set $\mathcal{D}_{\text{tr}}$ at training iteration $t$. An online batch selection algorithm aims to select the most valuable subset $\widehat{\mathcal{B}}_t$ from $\mathcal{B}_t$. It can be naturally formulated as an optimization problem, where the objective is to maximize the utility of the selected $\widehat{\mathcal{B}}_t$ for model update. Here, we describe the existing online batch selection algorithms through the concept of a *utility function*.

**Utility Function.** At training iteration $t$, a *utility function* $U^{(t)}$ maps an input training data batch $S$ to a score indicating the utility of this batch for the model update at iteration $t$. Specifically, for a given utility function $U$, the task of *online batch selection* over a candidate batch $\mathcal{B}_t$ is to identify the subset $\widehat{\mathcal{B}}_t \subseteq \mathcal{B}_t$ that optimizes:

$$\widehat{\mathcal{B}}_t^{(k)} = \underset{S \subseteq \mathcal{B}_t, |S|=k}{\operatorname{argmax}} U^{(t)}(S) \tag{1}$$

where $k$ is a fixed budget of sample number $k < |\mathcal{B}_t|$ used to update the model. Since $U^{(t)}$ is a set function, solving Equation (1) presents significant challenges, as it may require evaluating the utility $U^{(t)}(S)$ for a large number of subsets $S \subseteq \mathcal{B}_t$. Existing online batch selection methods circumvent this issue through *"Scoring and Top-k Paradigm"*, which compute an importance score $\phi_z$ for each data point $z \in \mathcal{B}_t$ and then selecting the subset of data points with the highest importance scores. For example, [28, 20] use the individual loss on the training data point $\phi_z = \ell(w_t, z)$ as the importance

---

[1]Here we incorporate the normalization term $|S|$ into the learning rate $\eta_t$ for notational simplicity.

score. [23] use the individual gradient norm $\phi_z = \|\nabla \ell(w_t, z)\|$ as the importance score. [30, 9] leverage a reference model and use the "reducible loss" as the importance score. The use of importance scores for online batch selection essentially defines the utility function $U^{(t)}(S) = \sum_{i \in S} \phi_i(U)$ and conjectures that the sum of individual data points' importance scores is a reliable indicator of a dataset $S$'s utility, hoping for a positive correlation with the model $w_t$'s performance at the $(t+1)$-th step after updating on $S$. Consequently, existing online batch selection strategies aim to maximize $\sum_{i \in S} \phi_i(U)$ by selecting the top-$k$ data points with the highest importance scores.

**Limitations of Scoring and Top-$k$ Paradigm.** Most scoring mechanisms for estimating the value of an individual data point $z \in \mathcal{B}_t$ result in similar data receiving similar scores. However, in the context of online batch selection, diversity is crucial. Consequently, a subset $\widehat{\mathcal{B}}_t$ consisting of the top-$K$ valued data points may lack diversity. In particular, duplicate points might be scored equally high and be incorrectly assumed to doubly improve the model, though this is likely not the case. The primary issue with this top-$K$ methodology is that it ignores the interactions among the selected data points. **When a data point is selected, the importance scores of the remaining data points will usually change.** For instance, the values of data points similar to the selected ones will typically decrease, while the values of data points that are very different from the selected ones will increase.

# 4 Optimizing Utility in Online Batch Selection via Greedy Algorithms

## 4.1 A Principled Utility Function for Online Batch Selection.

The performance of a model is typically measured through a set of validation points $\{z^{(\mathrm{val})}\}$. For a given validation data point $z^{(\mathrm{val})}$, an ideal utility function at a single iteration $t$ is the reduction in validation loss:

$$U^{(t)}(S; z^{(\mathrm{val})}) := \ell(w_t, z^{(\mathrm{val})}) - \ell(\widetilde{w}_{t+1}(S), z^{(\mathrm{val})}) \tag{2}$$

where $\widetilde{w}_{t+1}(S) := w_t - \eta_t \sum_{z \in S} \nabla \ell(w_t, z)$ and $S \subseteq \mathcal{B}_t$ is the subset of the batch selected for model update. While this is a principled choice for an optimization objective in online batch selection, optimizing $U^{(t)}$ is computationally expensive, as it involves evaluating model updates with respect to combinatorially many subsets $S \subseteq \mathcal{B}_t$ (a total of $\binom{|\mathcal{B}_t|}{k}$ subsets).

**Vanilla Greedy Algorithm.** To address the challenge of evaluating the objective function for numerous subsets, the greedy optimization algorithm is widely used due to its effectiveness in set function optimization. The greedy algorithm iteratively selects the element that provides the largest *marginal gain* to the utility function, given the previously selected elements. Mathematically, when a utility function $U^{(t)}$ is given, the greedy algorithm selects data points $z \in \mathcal{B}_t$ one at a time. At each selection round, the algorithm selects the data point $z^* = \mathrm{argmax}_{z \in \mathcal{B}_t \setminus \widehat{\mathcal{B}}_t} U^{(t)}(\widehat{\mathcal{B}}_t \cup \{z\}) - U^{(t)}(\widehat{\mathcal{B}}_t)$. This process continues until $k$ data points have been added to $\widehat{\mathcal{B}}_t$. The greedy algorithm is known to provide near-optimal solutions for monotone submodular set functions, with a famous $(1 - 1/e)$-approximation guarantee [31]. The greedy algorithm only requires $O(k|\mathcal{B}_t|)$ evaluations of $U^{(t)}$, a significant improvement over the $\binom{|\mathcal{B}_t|}{k}$ evaluations required by the brute-force method.

However, optimizing $U^{(t)}$ using the greedy algorithm is still not practical for online batch selection. Each evaluation of the utility function $U^{(t)}(S)$ in (2) involves computing aggregated gradients, updating the model, and calculating the validation loss, which can significantly increase the per-iteration cost of training. Since online batch selection algorithms run alongside the model training, these costs cannot be amortized across training runs as they would be with static dataset selection. This makes the greedy algorithm infeasible for real-time online batch selection.

## 4.2 An Efficient Greedy Algorithm for Utility Optimization without Utility Evaluation

Here, we develop an efficient approximation for the marginal gain of a data point $z \in \mathcal{B}_t \setminus \widehat{\mathcal{B}}_t$ to the utility of the already selected subset $\widehat{\mathcal{B}}_t$. For notational simplicity, we denote $\mathbf{g}_{(z_i)} := \nabla \ell(w_t, z_i)$ for all $z_i \in \mathcal{B}_t$. Given that the learning rate $\eta_t$ in model training is typically small, a lower-order Taylor expansion often provides an accurate approximation for the change in loss during a single gradient

---

**Algorithm 1** GREedy Approximation Taylor Selection (GREATS)

---

1: **Input:**
   - `HessianApprox`: approximation method for Hessian matrix.
2: Initialize model $w_0$.
3: **for** $t = 0, \ldots, T - 1$ **do**
4:     Sampled a random batch $\mathcal{B}_t$ from $\mathcal{D}_{\text{tr}}$.
5:     Initialize the importance score $\phi_z \leftarrow \eta_t \mathbf{g}_{(z)} \cdot \mathbf{g}_{(z^{(\text{val})})}$ for every $z \in \mathcal{B}_t$.
6:     Initialize the selected batch $\widehat{\mathcal{B}}_t \leftarrow \{\}$.
7:     **for** $r = 1, \ldots, k$ **do**
8:         $z^* \leftarrow \operatorname{argmax}_{z \in \mathcal{B}_t \setminus \widehat{\mathcal{B}}_t} \phi_z$.
9:         Add $z^*$ into $\widehat{\mathcal{B}}_t$.
10:        **if** `HessianApprox` = "exact" **then**
11:           Update the importance score $\phi_z \leftarrow \phi_z - \eta_t^2 \mathbf{g}_{(z)} \mathbf{H}_{(z^{(\text{val})})} \mathbf{g}_{(z^*)}$ for $z \in \mathcal{B}_t \setminus \widehat{\mathcal{B}}_t$.
12:        **else if** `HessianApprox` = "identity" **then**
13:           Update the importance score $\phi_z \leftarrow \phi_z - \eta_t^2 \mathbf{g}_{(z)} \cdot \mathbf{g}_{(z^*)}$ for $z \in \mathcal{B}_t \setminus \widehat{\mathcal{B}}_t$.
14:     Take one step of gradient update on the selected batch $\widehat{\mathcal{B}}_t$ and obtain $w_{t+1}$.

---

update, with approximation errors of $O(\eta_t^2)$ for first-order approximations.

$$
\begin{aligned}
U^{(t)}(z_i | \widehat{\mathcal{B}}_t) &:= U^{(t)}(\widehat{\mathcal{B}}_t \cup \{z_i\}) - U^{(t)}(\widehat{\mathcal{B}}_t) \\
&= \ell(\widetilde{w}_{t+1}(\widehat{\mathcal{B}}_t), z^{(\text{val})}) - \ell(\widetilde{w}_{t+1}(\widehat{\mathcal{B}}_t \cup \{z_i\}), z^{(\text{val})}) \\
&= \ell(\widetilde{w}_{t+1}(\widehat{\mathcal{B}}_t), z^{(\text{val})}) - \ell(\widetilde{w}_{t+1}(\widehat{\mathcal{B}}_t) - \eta_t \mathbf{g}_{(z_i)}, z^{(\text{val})}) \\
&\approx \eta_t \mathbf{g}_{(z_i)} \cdot \nabla \ell(\widetilde{w}_{t+1}(\widehat{\mathcal{B}}_t), z^{(\text{val})})
\end{aligned}
\tag{3}
$$

**Interpretation.** The first-order approximation of the marginal gain $U^{(t)}(z_i | \widehat{\mathcal{B}}_t)$ computes the inner-product between **(1)** the gradient of the individual training loss with respect to the original model $w_t$, and **(2)** the gradient of the validation loss with respect to the "virtual model" $\widetilde{w}_{t+1}(\widehat{\mathcal{B}}_t)$, i.e., $w_t$ updated with the existing selected batch. The inner-product represents the direct influence of $z_i$ on the validation loss at the "virtual model" $\widetilde{w}_{t+1}(\widehat{\mathcal{B}}_t)$. The gradient $\mathbf{g}_{(z_i)}$ is computed with respect to $w_t$ instead of $\widetilde{w}_{t+1}(\widehat{\mathcal{B}}_t)$ because the model update process is performed using the gradients with respect to $w_t$. The approximation in (3) essentially estimates the improvement in validation loss by including $z_i$ in the model update, assuming that $\widehat{\mathcal{B}}_t$ is already guaranteed to be included.

However, the computation of $\nabla \ell(\widetilde{w}_{t+1}(\widehat{\mathcal{B}}_t), z^{(\text{val})})$ still requires obtaining the updated model parameter $\widetilde{w}_{t+1}(\widehat{\mathcal{B}}_t)$ and performing additional backpropagations to compute the validation gradient, which again incurs significant computational overhead. To efficiently approximate it, we use another Taylor expansion as follows:

$$
\nabla \ell\left(\widetilde{w}_{t+1}(\widehat{\mathcal{B}}_t), z^{(\text{val})}\right) = \nabla \ell(w_t - \eta_t \sum_{z \in \widehat{\mathcal{B}}_t} \mathbf{g}_{(z)}, z^{(\text{val})}) \approx \mathbf{g}_{(z^{(\text{val})})} - \eta_t \mathbf{H}_{(z^{(\text{val})})} \sum_{z \in \widehat{\mathcal{B}}_t} \mathbf{g}_{(z)}
$$

where the Hessian matrix $\mathbf{H}_{(z^{(\text{val})})} := \nabla^2 \ell(w_t, z^{(\text{val})})$, and $\mathbf{g}_{(z^{(\text{val})})} := \nabla \ell(w_t, z^{(\text{val})})$. Plugging this approximation back into (3), we have

$$
U^{(t)}(z_i | \widehat{\mathcal{B}}_t) \approx \underbrace{\eta_t \mathbf{g}_{(z_i)} \cdot \mathbf{g}_{(z^{(\text{val})})}}_{\text{Importance score of } z_i} - \underbrace{\eta_t^2 \mathbf{g}_{(z_i)} \mathbf{H}_{(z^{(\text{val})})} \sum_{z \in \widehat{\mathcal{B}}_t} \mathbf{g}_{(z)}}_{\text{Correction of importance from already selected points}}
\tag{4}
$$

**Interpretation.** The first gradient inner-product $\eta_t \mathbf{g}_{(z_i)} \cdot \mathbf{g}_{(z^{(\text{val})})}$ coincides with the TracIN score proposed in [33] as a measure for a data point's importance. It captures the alignment between the gradient of the training loss for $z_i$ and the gradient of the validation loss, indicating how much the update based on $z_i$ would contribute to the reduction of the validation loss with respect to the original model $w_t$. The second term $-\eta_t^2 \mathbf{g}_{(z_i)} \mathbf{H}_{(z^{(\text{val})})} \sum_{z \in \widehat{\mathcal{B}}_t} \mathbf{g}_{(z)}$ is a correction term for $z_i$'s original importance after picking $\widehat{\mathcal{B}}_t$. It penalizes the similarity between $z_i$ and the data points in $\widehat{\mathcal{B}}_t$, as

measured by the Hessian-weighted inner-product of their gradients. Intuitively, if the gradient of $z_i$ is similar to the gradients of the data points in $\widehat{\mathcal{B}}_t$, the correction term will be large, reducing the overall marginal gain of adding $z_i$ to the selected subset. This encourages the selection of diverse data points that provide complementary information to the model update.

**Algorithm.** Using the approximation from (4), we develop a new algorithm that approximates the vanilla greedy algorithm. Initially, each data point $z \in \mathcal{B}_t$ is assigned an importance score $\phi_z$ initialized as $\phi_z = \eta_t \mathbf{g}_{(z)} \cdot \mathbf{g}_{(z^{(\text{val})})}$, which approximates the marginal gain of adding $z$ to an empty set, i.e., $U^{(t)}(z_i \mid \{\}) = U^{(t)}(\{z_i\})$. The algorithm begins by selecting the data point with the highest importance score, $z_1^* = \arg\max_{z \in \mathcal{B}_t} \phi_z$. After selecting a data point $z^*$ for model update, the importance scores for the remaining data points are adjusted by $-\eta_t^2 \mathbf{g}_{(z_i)} \mathbf{H}_{(z^{(\text{val})})} \mathbf{g}_{(z^*)}$. This adjustment approximates the marginal gain of adding each remaining data point to the set containing $z^*$, i.e., $U^{(t)}(z_i \mid \{z^*\})$. The algorithm iteratively selects the data point with the highest adjusted importance score and updates the scores for the remaining points until $k$ data points have been selected. As we can see, this iterative process closely mimics the behavior of the vanilla greedy algorithm while not requiring any actual evaluation of $U^{(t)}$, allowing for a computationally tractable approximation of the greedy algorithm in the context of online batch selection. The pseudocode for the proposed algorithm is detailed in Algorithm 1.

**Validity of Taylor Approximation.** We evaluate the fidelity of using Taylor expansion to approximate $U^{(t)}$. Following the experimental settings from our GPT2 experiments detailed in Appendix B, we sample different batch subsets $S$ and evaluate $U^{(t)}(S)$ at the 3500th training iteration. Figure 1 illustrates the correlation between actual and predicted validation loss changes in a single gradient update step. Panel (a) shows the correlation when using only the first-order term (the sum of training gradients' dot products with the validation point) for loss change approximation. Panel (b) demonstrates the improved correlation when incorporating both the first-order term and the Hessian interaction term. The enhanced correlation coefficient with the inclusion of the Hessian term indicates that our approximations effectively capture the actual loss dynamics, with the second-order term providing substantial improvement in predictive accuracy.

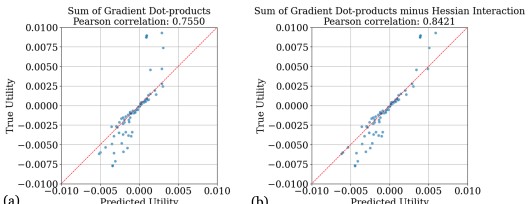

Figure 1: **(a)** We show the correlation between the ground-truth model validation loss change in one gradient update iteration $U^{(t)}(S; z^{(\text{val})}) := \ell(w_t, z^{(\text{val})}) - \ell(\widetilde{w}_{t+1}(S), z^{(\text{val})})$ and the first-order Taylor approximation $\sum_{z \in S} \eta_t \mathbf{g}_{(z)} \cdot \mathbf{g}_{(z^{(\text{val})})}$. **(b)** We show the correlation between $U^{(t)}(S; z^{(\text{val})})$ and the first-order approximation corrected by the Hessian interaction $\sum_{z \in S} \eta_t \mathbf{g}_{(z)} \cdot \mathbf{g}_{(z^{(\text{val})})} - \sum_{z,z' \in S} \eta_t^2 \mathbf{g}_{(z)} \mathbf{H}_{(z^{(\text{val})})} \mathbf{g}_{(z')}$.

## 4.3 The Ghost Inner-Product Technique

**Implementation Challenges of Algorithm 1.** Although Algorithm 1 eliminates the need for explicit utility evaluations and relies solely on gradient and Hessian information, its efficient implementation remains a challenge. The initial importance scores $\eta_t \mathbf{g}_{(z)} \cdot \mathbf{g}_{(z^{(\text{val})})}$ require computing the inner-products between the gradients of each $z \in \mathcal{B}_t$ and the validation point $z^{(\text{val})}$. Directly implementing this would involve calculating the individual gradients for every data point in $\mathcal{B}_t$. This cannot leverage the parallel processing capabilities of GPUs and would require running backpropagation $|\mathcal{B}_t|$ times with a mini-batch size of 1, resulting in a significantly higher per-iteration runtime cost compared to regular training. Furthermore, the correction term $-\eta_t^2 \mathbf{g}_{(z)} \mathbf{H}_{(z^{(\text{val})})} \mathbf{g}_{(z^*)}$ requires computing the gradient-Hessian-gradient product for each pair of $z, z' \in \mathcal{B}_t$. Even if we approximate the Hessian as the identity matrix ($\mathbf{H}_{(z^{(\text{val})})} \approx \mathbf{I}$), calculating the pairwise $\mathbf{g}_{(z)} \cdot \mathbf{g}_{(z')}$ still necessitates either storing all individual gradient vectors $\{\mathbf{g}_{(z)}\}$ or recomputing $\mathbf{g}_{(z)}$ during each round of greedy selection. Both the memory and computational demands of these approaches are impractical for training large-scale models.

**Efficient Computation of ALL Initial Importance Scores.** To address the challenge of computing the gradient inner-products between all $z \in \mathcal{B}_t$ and $z^{(\text{val})}$, we propose a novel technique called "ghost inner-product", which is inspired by the "ghost clipping" technique from the differential

privacy literature [5, 6]. The key idea behind "ghost inner-product" is to avoid explicitly computing individual gradient vectors, thereby improving the efficiency of the algorithm. To illustrate this technique, consider a simple linear layer $\mathbf{s} = \mathbf{aW}$, where $\mathbf{W} \in \mathbb{R}^{d_1 \times d_2}$ is the weight matrix, $\mathbf{a} = (\mathbf{a}^{(1)}, \ldots, \mathbf{a}^{(B)})^\mathsf{T} \in \mathbb{R}^{B \times d_1}$ is the mini-batch input, and $\mathbf{s} = (\mathbf{s}^{(1)}, \ldots, \mathbf{s}^{(B)})^\mathsf{T} \in \mathbb{R}^{B \times d_2}$ is the output (i.e., the pre-activation tensor). Let $\ell^{(i)}$ denote the individual loss on $z_i$. By applying the chain rule, we can express the gradient of an individual loss $\ell^{(i)} := \ell(w, z_i)$ with respect to $\mathbf{W}$ as

$$\frac{\partial \ell^{(i)}}{\partial \mathbf{W}} = \frac{\partial \ell^{(i)}}{\partial \mathbf{s}^{(i)}} \frac{\partial \mathbf{s}^{(i)}}{\partial \mathbf{W}} = \frac{\partial \ell}{\partial \mathbf{s}^{(i)}} \mathbf{a}^{(i)} \tag{5}$$

where $\ell := \sum_{j=1}^{B} \ell^{(j)}$ is the aggregated loss. Note that the output gradient $\frac{\partial \ell}{\partial \mathbf{s}^{(i)}}$ is readily available during the backpropagation pass. To efficiently compute the gradient inner-product between a validation point and each training point, we include the validation data $z^{(\text{val})}$ together in the batch for backpropagation. That is, we take the backpropagation on $\sum_{j=1}^{B} \ell^{(j)} + \ell^{(z^{(\text{val})})}$. Hence, for each training-validation pair $(z_i, z^{(\text{val})})$, we have the gradient inner-product

$$\frac{\partial \ell^{(i)}}{\partial \mathbf{W}} \odot \frac{\partial \ell^{(z^{(\text{val})})}}{\partial \mathbf{W}} = \left( \left( \frac{\partial \ell}{\partial \mathbf{s}^{(i)}} \right)^\mathsf{T} \left( \frac{\partial \ell}{\partial \mathbf{s}^{(z^{(\text{val})})}} \right) \right) \left( \left( \mathbf{a}^{(i)} \right)^\mathsf{T} \mathbf{a}^{(z^{(\text{val})})} \right)$$

By using the "ghost inner-product" technique, we can compute the result without explicitly forming any full gradient vectors. Consequently, computing the gradient inner-product between every pair of training and validation points requires only one backpropagation, which is significantly more efficient than the direct method that would require $> |\mathcal{B}_t|$ backpropagations. We note that the "ghost inner-product" technique can be applied to various types of layers beyond linear layers. Similar decompositions as in Equation (5) have been studied in the differential privacy literature [36, 5, 26], enabling the extension of this technique to other layer types. Extension on LoRA is in Appendix A.

**Efficient Approximation of Importance Correction.** The importance correction term $\mathbf{g}_{(z)} \mathbf{H}_{(z^{(\text{val})})} \mathbf{g}_{(z^*)}$ poses computational challenges due to the involvement of the Hessian matrix. A straightforward approximation is to assume $\mathbf{H}_{(z^{(\text{val})})} \approx \mathbf{I}$, simplifying the problem to computing the gradient inner-product $\mathbf{g}_{(z)} \cdot \mathbf{g}_{(z^*)}$. This approximation has been widely used in the literature, particularly in the context of second-order optimization methods and meta-learning [29, 13, 32]. The key motivation behind this approximation is that the Hessian matrix is often diagonally dominant, especially when the model is close to a local minimum [4]. By assuming $\mathbf{H}_{(z^{(\text{val})})} \approx \mathbf{I}$, the importance correction term simplifies to $\phi_z - \eta_t^2 \mathbf{g}_{(z)} \cdot \mathbf{g}_{(z^*)}$. We can then apply the ghost inner-product technique previously developed for computing pairwise gradient inner-products. This allows us to efficiently compute the importance correction term without explicitly forming the individual gradient vectors or the Hessian-vector products.

**Merging Batch Selection and Gradient Update in One Backpropagation.** Utilizing the techniques developed in this section, we can calculate or approximate all importance scores and correction terms in a single backpropagation pass, without the need to materialize any model-sized vectors. Although computing the gradient of the aggregated training loss $\sum_{z_i \in \mathcal{B}_t} \ell^{(i)}$ for the training batch is necessary for parameter updates, an additional backpropagation pass is not required. By retaining the activations and output gradients from the previous backpropagation, we can efficiently compute this gradient without incurring the cost of another pass (see Appendix A.3 for details). Consequently, the process of training with batch selection introduces minimal additional runtime overhead. This approach provides substantial benefits over the direct method of materializing per-sample model-sized gradients, making it more feasible for real-world applications.

## 5 Experiments

In this section, we first evaluate the performance of GREATS against several baselines on a diverse set of models, training datasets, and validation set configurations. We then empirically examine its computational efficiency when implemented with "ghost inner-product" technique from Section 4.3.

### 5.1 Experimental Setup

**Model-Training-Evaluation Pairs.** We examine multiple combinations of models, training datasets, and evaluation datasets to evaluate our proposed GREATS algorithm, as shown in Table 1. Specifically,

Table 1: Combination of models, training datasets, and evaluation datasets

| Task | Model | Training Dataset | Evaluation Dataset | Number of validation data |
|------|-------|------------------|--------------------|---------------------------|
| Fine-tuning | LLAMA-2-7B [38] | LESS [43] | MMLU [17] | 5 |
| Fine-tuning | MISTRAL-7B [19] | LESS [43] | TYDIQA [7] | 10 |
| Fine-tuning | LLAMA-3-8B [1] | ALPACA [37] | SAMSUM [14] | 16 |
| Pretraining | GPT-SMALL [34] | OPENWEBTEXT [15] | OPENWEBTEXT [15] | 16 |

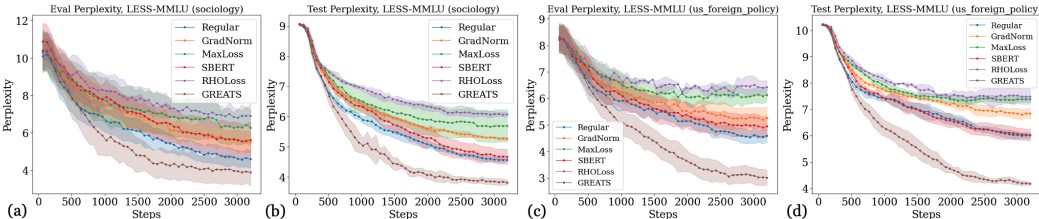

Figure 2: Comparison of the validation and test perplexity dynamics during training for different online batch selection methods on MMLU. We select sociology and US foreign policy subjects.

we fine-tune three large language models (LLMs): LLAMA-2-7B [38], MISTRAL-7B [19], and LLAMA-3-8B [1] using LESS training data [43] and the ALPACA dataset [37]. For evaluation, we employ the MMLU [17], TYDIQA [7], and SAMSUM [14] datasets (deferred to Appendix C). Additionally, we conduct a pretraining experiment using the GPT-SMALL model [34]. For both training and evaluation, we use the OPENWEBTEXT dataset [15]. In all experiments, we limited the validation data to be small (i.e., $\leq 16$) to mimic practical scenarios where training directly on them is impossible.

**Baselines.** We compare our algorithm with regular training and a variety of online batch selection algorithms: **(1)** MaxLoss [28], which selects training data points with the highest loss values. **(2)** GradNorm [23], which prioritizes training data points with the highest gradient norms. **(3)** Reference model-based method (RHOLoss), for which we implement the RHO-Loss algorithm from [30] as a representative baseline. Given computational constraints, we use Llama-3.1-8B-Instruct [10] as the reference model, paired with Llama2-7B as the target model.[2] **(4)** Distance-based method (SBERT), which selects training batches based on their semantic similarity to validation data using Sentence-BERT embeddings [35].

**Training Details.** For all batch selection methods, we select 50% of the batch data for gradient updates during each step. In contrast, the regular training baseline performs updates on the entire batch, utilizing *twice* as much data as the batch selection methods. In the main paper, we show the results of setting the batch size to 4 for MMLU and TYDIQA, 16 for SAMSUM and OPENWEBTEXT. Additional training details and ablation studies are provided in Appendix B.

## 5.2 Performance Evaluation

In this section, we present and discuss the comparison between GREATS and the baseline algorithms in model training performance across different models, training, and evaluation datasets. We evaluate the performance on both the validation set (being used for batch selection in GREATS) and a test set that is drawn from the same domain of the evaluation datasets.

**GREATS significantly speeds up training convergence.** In Figure 2 and 4, we show the dynamics of perplexity on the validation and test datasets. As we can see from Figure 2 and Figure 4, across all settings, the GREATS algorithm achieves a significantly faster reduction in test perplexity compared to all of the baselines and often achieves better overall performance.

---

[2]We note that while larger pretrained models could serve as reference models, their computational costs increase substantially, particularly due to the need for repeated queries at each training iteration. Furthermore, in specialized domains, general-purpose pretrained models may exhibit suboptimal performance, necessitating fine-tuning on substantial holdout datasets. This additional requirement becomes impractical when computational efficiency and data availability are crucial considerations. We thank the anonymous NeurIPS reviewer for their helpful suggestion regarding the reference model selection in our comparisons.

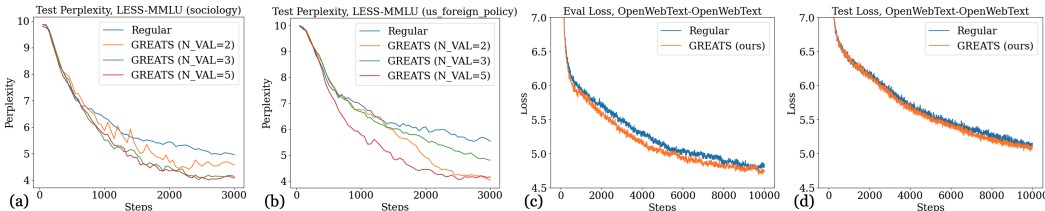

Figure 3: (a)-(b): Impact of the number of validation data points on the performance of GREATS. (c)-(d): Comparison of the validation and test perplexity dynamics during GPT2 pretraining for different online batch selection methods.

| Method | MMLU (AVG.) | Soc. | Pol. | Alg. | Anat. | Astr. | Eth. | Clin. | Bio. | Chem. | TYDIQA |
|---|---|---|---|---|---|---|---|---|---|---|---|
| **Regular** | 50.4% | 62% | 60% | 40% | 52% | 48% | 52% | **54%** | 48% | 38% | 54.3% |
| **GradNorm** | 50.4% | 62% | 62% | 38% | 50% | 48% | 54% | 52% | **50%** | 38% | 53.4% |
| **MaxLoss** | 50.8% | 64% | 58% | 40% | 52% | 46% | 54% | **54%** | **50%** | 40% | 54.7% |
| **Ours** | **54.2%** | **68%** | **64%** | **44%** | **56%** | **52%** | **56%** | **54%** | **50%** | **44%** | **55.0%** |

Table 2: Accuracy on MMLU (9 subjects) and TYDIQA test set for online batch selection methods.

These results demonstrate the robust effectiveness of our approach in improving model convergence speed and generalization performance across various settings. We note that the validation-free approaches such as `GradNorm` and `MaxLoss` may lead to the selection of low-quality data. While it is generally considered that data points with high training loss or large gradients are important to learning, there is another possibility that the data points that achieve these properties are corrupted data which is not learnable.

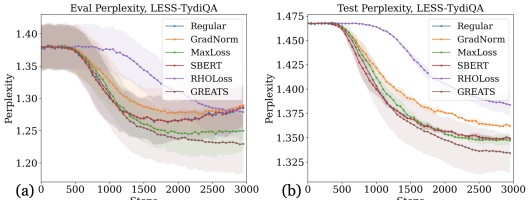

Figure 4: Comparison of the validation and test perplexity dynamics during training for different online batch selection methods on TYDIQA.

**GREATS improves performance on downstream tasks.** While perplexity is a direct measure of the performance of NLP models, it is not very interpretable, and the performance is often evaluated in terms of downstream tasks. In Table 2, we show the test accuracy on 9 (randomly selected) subjects from MMLU and TYDIQA. As we can see, GREATS consistently outperforms or at least achieves the same accuracy as the baselines. Notably, we observe at least a 3.4% improvement in the average performance of the 9 MMLU subjects compared to all baselines.

**GREATS is robust to the number of validation points.** In Figure 3 (a)-(b), we conduct an ablation study to evaluate the impact of the number of validation points used for GREATS algorithm. As we can see, even with just 2 validation examples, the test perplexity on the MMLU dataset is consistently lower than that of regular training. This robustness may be attributed to the relatively rare format of the validation corpus, which allows GREATS to effectively select examples from the batch that can help learn the particular format. Even if such a selection may overfit the specific validation examples, the selected batch can still improve the performance on the test examples, demonstrating the effectiveness of the GREATS algorithm in adapting to the characteristics of the validation set.

**GREATS also improves LLM pretraining.** In Figure 3 (c)-(d), we evaluate the performance of GREATS on pretraining GPT-SMALL on OPENWEBTEXT. Due to computational resource constraints, we omit the baselines of `GradNorm` and `MaxLoss`, as they have been shown to be ineffective in all other experiments. As we can see, even for model pretraining, GREATS provides an improvement in test performance, although the improvement is marginal compared to the gains observed in the fine-tuning experiments. This marginal improvement can be attributed to the fact that the validation data used in this experiment is also drawn from the same distribution as the training set, i.e., OPENWEBTEXT. As a result, the selected batches may not provide as much additional information or diversity as in the case of fine-tuning, where the validation data often comes from a different distribution or focuses on specific tasks. Nevertheless, the consistent improvement in test performance suggests that GREATS can still identify informative examples that contribute to better model generalization, even in the pretraining setting.

## 5.3 Runtime Comparison

We compare the runtime efficiency of GREATS algorithm with "ghost inner-product" technique against GREATS implemented directly by calculating per-sample gradients. The runtime is measured by training GPT-SMALL on OPENWEBTEXT. Additionally, we compare against `GradNorm`'s direct implementation using per-sample gradients, as it is the most similar algorithm to GREATS and allows for a fair comparison.

**GREATS with "ghost inner-product" achieves runtime close to regular training.** As shown in Table 3, the runtime of GREATS using our efficient approximation techniques is comparable to that of regular training, with only a slight increase in runtime due to pairwise inner-product operations (but almost negligible compared with model backpropagation). This demonstrates the effectiveness of our approximation methods in reducing the computational overhead associated with online batch selection. On the other hand, the direct implementation of GREATS, which requires computing per-sample gradients, exhibits a significant runtime increase compared to both regular training and our efficient GREATS implementation. The direct approach is significantly slower than regular training (almost 20 times slower), making it impractical for real-world applications.

|  | Throughput |
|---|---|
| **Regular Training** | 76.2 |
| **GREATS (ghost)** | 71.3 |
| **GREATS (direct)** | 4.2 |
| **GradNorm (direct)** | 6.8 |

Table 3: Efficiency comparison of different implementations of GREATS. We use throughput-# training data points being processed per second-as the efficiency metric.

The runtime of `GradNorm` with direct per-sample gradient computation falls between our efficient GREATS implementation and the direct GREATS implementation. This is because `GradNorm` does not need to compute the per-sample gradients from the validation set, which reduces its computational overhead compared to the direct GREATS implementation. However, `GradNorm` still incurs a significant runtime increase compared to regular training due to the per-sample gradient calculation for the training batch. We remark that the "ghost norm" technique from differential privacy literature [16, 25, 5, 6], which is similar to "ghost inner-product", can be used to improve the runtime of `GradNorm`, potentially bringing it closer to regular training.

## 6 Conclusion and Limitations

In this work, we introduced GREATS, a novel online batch selection algorithm designed to enhance the efficiency and effectiveness of training large language models. Here, we briefly summarize the limitations of this work.

**I. Availability of validation data.** One potential limitation of GREATS is that it requires the validation data to be available before training. We stress that there are many scenarios where the validation data is naturally available before training such as fine-tuning or domain adaptation. Developing a validation-free variant of GREATS is an interesting future work.

**II. Extension to Adam.** The ghost inner-product technique developed in this work is specifically tailored for Stochastic Gradient Descent (SGD). It is not directly extendable to other popular optimizers like Adam due to their normalization terms. Nonetheless, using SGD as a proxy for Adam has proved to be effective in our experiment. Extending our ghost inner-product technique to Adam and similar optimizers remains an exciting direction for future research.

**III. Memory constraint for large batch sizes.** In scenarios where GPU memory constraints prevent adding validation data to the training batch for backpropagation, we can easily extend our "ghost" techniques by using gradient accumulation. However, this may increase runtime due to additional backpropagation steps for validation data, it maintains the feasibility of our techniques under memory constraints. Improving computational efficiency for large batch sizes remains an important direction for future research.

**IV. Perplexity may not be an ideal objective.** In this work, the utility function is being defined as the validation loss. While GREATS achieves promising results overall in terms of test perplexity, we note that perplexity may not reflect the performance in downstream tasks. While GREATS usually achieves higher performance on the downstream task, the improvement is often minor. Directly optimizing in terms of the downstream performances is another important future work.

## Acknowledgment

This work is supported in part by the National Science Foundation under grants IIS-2312794, IIS-2313130, OAC-2239622, CNS-2131938, Amazon-Virginia Tech Initiative in Efficient and Robust Machine Learning, the Commonwealth Cyber Initiative, OpenAI and Google.

We thank Ashwinee Panda, Xinyu Tang, and Yiding Jiang for their helpful feedback on the preliminary version of this work. We thank anonymous NeurIPS reviewers for the helpful feedback and discussion of this work.

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

# A  Ghost Inner-Product

**Notation Review.** Consider a linear layer $\mathbf{s} = \mathbf{a}\mathbf{W}$, where $\mathbf{W} \in \mathbb{R}^{d_1 \times d_2}$ is the weight matrix, $\mathbf{a} = (\mathbf{a}^{(1)}, \dots, \mathbf{a}^{(B)})^\mathsf{T}$ is the mini-batch input, and $\mathbf{s} = (\mathbf{s}^{(1)}, \dots, \mathbf{s}^{(B)})^\mathsf{T}$ is the layer output (i.e., the pre-activation tensor). For non-sequential data, $\mathbf{a} \in \mathbb{R}^{B \times d_1}, \mathbf{s} \in \mathbb{R}^{B \times d_2}$. For sequential data with sequence length $T$, $\mathbf{a} \in \mathbb{R}^{B \times d_1 \times T}, \mathbf{s} \in \mathbb{R}^{B \times d_2 \times T}$. Let $\ell^{(i)} := \ell(w, z_i)$ denote the current model's individual loss on $z_i$, and we denote $\ell := \sum_{j=1}^{B} \ell^{(j)}$ the aggregated loss over the full data batch $\{z_j\}_{j=1}^{B}$. For notation convenience, we denote individual output derivative $\mathbf{b}^{(i)} := \left( \frac{\partial \ell^{(i)}}{\partial \mathbf{s}^{(i)}} \right)^\mathsf{T}$.

## A.1  Ghost Inner-Product for Linear Layers

**Non-sequential data.** For non-sequential data, we can decompose the gradient of an individual loss $\ell^{(i)}$ with respect to $\mathbf{W}$ as

$$\frac{\partial \ell^{(i)}}{\partial \mathbf{W}} = \frac{\partial \ell^{(i)}}{\partial \mathbf{s}^{(i)}} \frac{\partial \mathbf{s}^{(i)}}{\partial \mathbf{W}} = \frac{\partial \ell}{\partial \mathbf{s}^{(i)}} \frac{\partial \mathbf{s}^{(i)}}{\partial \mathbf{W}} = \mathbf{a}^{(i)} \left( \frac{\partial \ell}{\partial \mathbf{s}^{(i)}} \right)^\mathsf{T} = \mathbf{a}^{(i)} \otimes \mathbf{b}^{(i)} \tag{6}$$

where the second equality is because $\ell^{(j)}$ does not depend on $\mathbf{s}^{(i)}$ for $j \neq i$, and the third equality is due to the linear transformation $\mathbf{s} = \mathbf{a}\mathbf{W}$. An important observation is that the individual's output gradient $\mathbf{b}^{(i)}$ is *readily available* during the backpropagation with respect to the aggregated loss $\ell$.

Say we are interested in computing the gradient inner-product $\frac{\partial \ell^{(1)}}{\partial \mathbf{W}} \odot \frac{\partial \ell^{(2)}}{\partial \mathbf{W}}$ between two data points $z_1, z_2$ in the same batch in the backpropagation. For non-sequential data, we have each $\mathbf{a}^{(i)} \in \mathbb{R}^{d_1 \times 1}$ and $\mathbf{b}^{(i)} \in \mathbb{R}^{1 \times d_2}$. By (6), we have

$$\frac{\partial \ell^{(1)}}{\partial \mathbf{W}} \odot \frac{\partial \ell^{(2)}}{\partial \mathbf{W}} = \left( \mathbf{a}^{(1)} \otimes \mathbf{b}^{(1)} \right) \odot \left( \mathbf{a}^{(2)} \otimes \mathbf{b}^{(2)} \right) = \left( \mathbf{b}^{(1)} \odot \mathbf{b}^{(2)} \right) \left( \mathbf{a}^{(1)} \odot \mathbf{a}^{(2)} \right) \tag{7}$$

where the second equality is due to the mixed product property. (7) is particularly interesting because it shows that we can compute the inner-product between $\frac{\partial \ell^{(1)}}{\partial \mathbf{W}}$ and $\frac{\partial \ell^{(2)}}{\partial \mathbf{W}}$ without actually instantiating the huge gradient vector $\frac{\partial \ell^{(1)}}{\partial \mathbf{W}}$ or $\frac{\partial \ell^{(2)}}{\partial \mathbf{W}}$. We can first take the dot products (i.e., inner-product for vectors) for $\mathbf{a}^{(1)} \odot \mathbf{a}^{(2)}$ and $\mathbf{b}^{(1)} \odot \mathbf{b}^{(2)}$, then multiply the results together. Moreover, all of the materials $\mathbf{a}^{(1)}, \mathbf{a}^{(2)}, \mathbf{b}^{(1)}, \mathbf{b}^{(2)}$ that are required for computation are all already available during the backpropagation with respect to $\ell$. Hence, with just a single backpropagation, we can efficiently compute the gradient inner-product between *every* pair of the data points within the batch.

**Sequential data.** For sequential data, we have each $\mathbf{a}^{(i)} \in \mathbb{R}^{d_1 \times T}$ and $\mathbf{b}^{(i)} \in \mathbb{R}^{T \times d_2}$. We can similarly decompose the gradient of an individual loss $\ell^{(i)}$ with respect to $\mathbf{W}$ as follows:

$$\frac{\partial \ell^{(i)}}{\partial \mathbf{W}} = \frac{\partial \ell^{(i)}}{\partial \mathbf{s}^{(i)}} \frac{\partial \mathbf{s}^{(i)}}{\partial \mathbf{W}} = \frac{\partial \ell}{\partial \mathbf{s}^{(i)}} \frac{\partial \mathbf{s}^{(i)}}{\partial \mathbf{W}} = \mathbf{a}^{(i)} \left( \frac{\partial \ell}{\partial \mathbf{s}^{(i)}} \right)^\mathsf{T} = \sum_{t=1}^{T} \mathbf{a}_t^{(i)} \otimes \mathbf{b}_t^{(i)} \tag{8}$$

By (8), we have

$$\frac{\partial \ell^{(1)}}{\partial \mathbf{W}} \odot \frac{\partial \ell^{(2)}}{\partial \mathbf{W}} = \left( \sum_{t=1}^{T} \mathbf{a}_t^{(1)} \otimes \mathbf{b}_t^{(1)} \right) \odot \left( \sum_{t=1}^{T} \mathbf{a}_t^{(2)} \otimes \mathbf{b}_t^{(2)} \right)$$

$$= \sum_{t_1=1}^{T} \sum_{t_2=1}^{T} \left( \mathbf{a}_{t_1}^{(1)} \otimes \mathbf{b}_{t_1}^{(1)} \right) \odot \left( \mathbf{a}_{t_2}^{(2)} \otimes \mathbf{b}_{t_2}^{(2)} \right)$$

$$= \sum_{t_1=1}^{T} \sum_{t_2=1}^{T} \sum_{j=1}^{d_1} \sum_{k=1}^{d_2} \left[ \mathbf{a}_{t_1,j}^{(1)} \mathbf{b}_{t_1,k}^{(1)} \right] \left[ \mathbf{a}_{t_2,j}^{(2)} \mathbf{b}_{t_2,k}^{(2)} \right]$$

$$= \sum_{t_1=1}^{T} \sum_{t_2=1}^{T} \left( \sum_{j=1}^{d_1} \mathbf{a}_{t_1,j}^{(1)} \mathbf{a}_{t_2,j}^{(2)} \right) \left( \sum_{k=1}^{d_2} \mathbf{b}_{t_1,k}^{(1)} \mathbf{b}_{t_2,k}^{(2)} \right)$$

$$= \sum_{t_1=1}^{T} \sum_{t_2=1}^{T} (\mathbf{a}_{t_1}^{(1)})^{\top} \mathbf{a}_{t_2}^{(2)} \, \mathbf{b}_{t_1}^{(1)} (\mathbf{b}_{t_2}^{(2)})^{\top}$$

$$= \left( \left( \mathbf{a}^{(1)} \right)^{\top} \mathbf{a}^{(2)} \right) \odot \left( \mathbf{b}^{(1)} \left( \mathbf{b}^{(2)} \right)^{\top} \right)$$

Hence, comparing with directly computing per-sample gradients, if $2T^2 < d_1 d_2$, it is more memory-efficient to first multiply the matrices of $\left( \mathbf{b}^{(1)} \right) \left( \mathbf{b}^{(2)} \right)^{\top}$ and $\left( \mathbf{a}^{(1)} \right)^{\top} \mathbf{a}^{(2)}$, then take the inner product between the two $T \times T$ matrices. If $2T^2 \geq d_1 d_2$, then we can first take the outer products $\mathbf{a}^{(1)} \otimes \mathbf{b}^{(1)}$ and $\mathbf{a}^{(2)} \otimes \mathbf{b}^{(2)}$, then take their inner product. In either case, we only need a single backpropagation to compute the gradient inner-product between every pair of the data points within the batch, similar to the case of non-sequential data.

Concurrent with this work, we also apply the ghost inner-product technique for efficient data attribution [39].

## A.2   Ghost Inner-Product for LoRA

**LoRA.** For a linear layer $\mathbf{W}$, LoRA (Low-Rank Adaptation) [18] introduces additional trainable parameters to adapt the model effectively while maintaining computational efficiency. Specifically, the original weight matrix $\mathbf{W}$ is modified as $\mathbf{W}' = \mathbf{W} + \mathbf{AB}$, where $\mathbf{A} \in \mathbb{R}^{d_1 \times r}$ and $\mathbf{B} \in \mathbb{R}^{r \times d_2}$ are new low-rank matrices with $r \ll \min(d_1, d_2)$. This adaptation allows for significant modifications of the layer's behavior through a low-rank update, which adds a minimal number of parameters to the model compared to the original number in $\mathbf{W}$. Here, we show how to compute ghost inner-product for LoRA.

The gradient of LoRA layer $\mathbf{A}$ can be computed as

$$\frac{\partial \ell}{\partial \mathbf{A}} = \frac{\partial \ell}{\partial \mathbf{W}'} \frac{\partial \mathbf{W}'}{\partial \mathbf{A}} = \left( \frac{\partial \ell}{\partial \mathbf{s}} \frac{\partial \mathbf{s}}{\partial \mathbf{W}'} \right) \mathbf{B}^{\mathsf{T}} = (\mathbf{a} \otimes \mathbf{b}) \, \mathbf{B}^{\mathsf{T}}$$

and the gradient of LoRA layer $\mathbf{B}$ can be computed as

$$\frac{\partial \ell}{\partial \mathbf{B}} = \frac{\partial \ell}{\partial \mathbf{W}'} \frac{\partial \mathbf{W}'}{\partial \mathbf{B}} = \mathbf{A}^{\mathsf{T}} (\mathbf{a} \otimes \mathbf{b})$$

**Non-sequential data.** For non-sequential data, the inner-product between two gradients on $\mathbf{A}$ can be written as

$$\frac{\partial \ell^{(1)}}{\partial \mathbf{A}} \cdot \frac{\partial \ell^{(2)}}{\partial \mathbf{A}} = \sum_{j=1}^{d_1} \sum_{k=1}^{r} \left( \left( \mathbf{a}^{(1)} \otimes \mathbf{b}^{(1)} \right) \mathbf{B}^\mathsf{T} \right)_{jk} \cdot \left( \left( \mathbf{a}^{(2)} \otimes \mathbf{b}^{(2)} \right) \mathbf{B}^\mathsf{T} \right)_{jk}$$

$$= \sum_{j=1}^{d_1} \sum_{k=1}^{r} \left( \mathbf{a}_j^{(1)} \sum_{i=1}^{d_2} \mathbf{b}_i^{(1)} \mathbf{B}_{ik}^\mathsf{T} \right) \left( \mathbf{a}_j^{(2)} \sum_{i=1}^{d_2} \mathbf{b}_i^{(2)} \mathbf{B}_{ik}^\mathsf{T} \right)$$

$$= \sum_{j=1}^{d_1} \left( \mathbf{a}_j^{(1)} \mathbf{a}_j^{(2)} \right) \sum_{k=1}^{r} \left( \sum_{i=1}^{d_2} \mathbf{b}_i^{(1)} \mathbf{B}_{ik}^\mathsf{T} \right) \left( \sum_{i=1}^{d_2} \mathbf{b}_i^{(2)} \mathbf{B}_{ik}^\mathsf{T} \right)$$

$$= \left( \mathbf{a}^{(1)} * \mathbf{a}^{(2)} \right) \left( \left( \mathbf{b}^{(1)} \mathbf{B}^\mathsf{T} \right) * \left( \mathbf{b}^{(2)} \mathbf{B}^\mathsf{T} \right) \right)$$

Similarly, we have

$$\frac{\partial \ell^{(1)}}{\partial \mathbf{B}} \cdot \frac{\partial \ell^{(2)}}{\partial \mathbf{B}} = \sum_{j=1}^{d_1} \sum_{k=1}^{r} \left( \mathbf{A}^\mathsf{T} \left( \mathbf{a}^{(1)} \otimes \mathbf{b}^{(1)} \right) \right)_{jk} \cdot \left( \mathbf{A}^\mathsf{T} \left( \mathbf{a}^{(2)} \otimes \mathbf{b}^{(2)} \right) \right)_{jk}$$

$$= \sum_{j=1}^{d_1} \sum_{k=1}^{r} \left( \sum_{i=1}^{d_2} \mathbf{A}_{ji}^\mathsf{T} \mathbf{a}_i^{(1)} \mathbf{b}_k^{(1)} \right) \left( \sum_{i=1}^{d_2} \mathbf{A}_{ji}^\mathsf{T} \mathbf{a}_i^{(2)} \mathbf{b}_k^{(2)} \right)$$

$$= \sum_{k=1}^{r} \left( \mathbf{b}_k^{(1)} \mathbf{b}_k^{(2)} \right) \sum_{j=1}^{d_1} \left( \sum_{i=1}^{d_2} \mathbf{A}_{ji}^\mathsf{T} \mathbf{a}_i^{(1)} \right) \left( \sum_{i=1}^{d_2} \mathbf{A}_{ji}^\mathsf{T} \mathbf{a}_i^{(2)} \right)$$

$$= \left( \mathbf{b}^{(1)} * \mathbf{b}^{(2)} \right) \left( \left( \mathbf{A}^\mathsf{T} \mathbf{a}^{(1)} \right) * \left( \mathbf{A}^\mathsf{T} \mathbf{a}^{(2)} \right) \right)$$

**Sequential data.** We now consider the setting of sequential data with sequence length $T$. In this case, we have $\mathbf{a} = (\mathbf{a}^{(1)}, \ldots, \mathbf{a}^{(B)})^\mathsf{T} \in \mathbb{R}^{B \times d_1 \times T}$ and $\mathbf{b} = (\mathbf{b}^{(1)}, \ldots, \mathbf{b}^{(B)})^\mathsf{T} \in \mathbb{R}^{B \times T \times d_2}$.

$$\frac{\partial \ell^{(1)}}{\partial \mathbf{A}} \cdot \frac{\partial \ell^{(2)}}{\partial \mathbf{A}} = \sum_{j=1}^{d_1} \sum_{k=1}^{r} \left( \sum_{i=1}^{d_2} \left( \sum_{t=1}^{T} \mathbf{a}_{jt}^{(1)} \mathbf{b}_{ti}^{(1)} \right) \mathbf{B}_{ik}^\mathsf{T} \right) \left( \sum_{i=1}^{d_2} \left( \sum_{t=1}^{T} \mathbf{a}_{jt}^{(2)} \mathbf{b}_{ti}^{(2)} \right) \mathbf{B}_{ik}^\mathsf{T} \right)$$

$$= \sum_{j=1}^{d_1} \sum_{k=1}^{r} \left( \sum_{t=1}^{T} \mathbf{a}_{jt}^{(1)} \sum_{i=1}^{d_2} \mathbf{b}_{ti}^{(1)} \mathbf{B}_{ik}^\mathsf{T} \right) \left( \sum_{t=1}^{T} \mathbf{a}_{jt}^{(2)} \sum_{i=1}^{d_2} \mathbf{b}_{ti}^{(2)} \mathbf{B}_{ik}^\mathsf{T} \right)$$

$$= \sum_{j=1}^{d_1} \sum_{k=1}^{r} \left( \sum_{t=1}^{T} \mathbf{a}_{jt}^{(1)} \mathbf{b}_{t,.}^{(1)} \mathbf{B}_{,.k}^\mathsf{T} \right) \left( \sum_{t=1}^{T} \mathbf{a}_{jt}^{(2)} \mathbf{b}_{t,.}^{(2)} \mathbf{B}_{,.k}^\mathsf{T} \right)$$

$$= \sum_{j=1}^{d_1} \sum_{k=1}^{r} \left( \sum_{t=1}^{T} \mathbf{a}_{jt}^{(1)} \mathbf{b}_{t,.}^{(1)} \mathbf{B}_{,.k}^\mathsf{T} \right) \left( \sum_{t=1}^{T} \mathbf{a}_{jt}^{(2)} \mathbf{b}_{t,.}^{(2)} \mathbf{B}_{,.k}^\mathsf{T} \right)$$

$$= \sum_{j=1}^{d_1} \sum_{k=1}^{r} \left( \sum_{t=1}^{T} \mathbf{a}_{jt}^{(1)} \mathbf{D}_{tk}^{(1)} \right) \left( \sum_{t=1}^{T} \mathbf{a}_{jt}^{(2)} \mathbf{D}_{tk}^{(2)} \right)$$

$$= \left( \mathbf{a}^{(1)} \right)^\mathsf{T} \left( \mathbf{a}^{(2)} \right) \cdot \left( \mathbf{D}^{(1)} \right) \left( \mathbf{D}^{(2)} \right)^\mathsf{T}$$

where in the second-to-the-last step we denote $\mathbf{D}_{tk}^{(2)} := \mathbf{b}_{t,.}^{(2)} \mathbf{B}_{,.k}^\mathsf{T}$.

### A.3 Merging Batch Selection and Gradient Update in One Backpropagation

By utilizing the ghost inner-product technique developed in this paper, we can calculate or approximate all importance scores and correction terms in a single backpropagation pass, without materializing any model-sized vectors. To compute the gradient inner-product between each training

point $z_i \in \mathcal{B}_t$ and the validation data $z^{(\text{val})}$, we propose including $z^{(\text{val})}$ in the backpropagation along with the training batch. Specifically, we can backpropagate with respect to $\sum_{z_i \in \mathcal{B}_t} \ell^{(i)} + \ell^{(z^{(\text{val})})}$.

After performing GREATS and selecting $\widehat{\mathcal{B}}_t$, it may seem necessary to backpropagate with respect to $\sum_{z_i \in \widehat{\mathcal{B}}_t} \ell^{(i)}$ to compute the gradient for the parameter update. However, this is not required. We can simply reuse the output gradient $\frac{\partial \ell^{(i)}}{\partial \mathbf{s}^{(i)}}$ from the original backpropagation and aggregate the gradients for all selected data points. This technique, referred to as the "book-keeping trick," is adapted from [6].

# B    Details of Experimental Setup

**Training Dataset.** In our experiments, we use three training datasets: LESS [43], ALPACA [37][CC-BY-NC 4.0], and OPENWEBTEXT [15][CC0]. Specifically, LESS is a combination of four instruction tuning datasets: FLAN V2 [27], COT [41], DOLLY [8], and OPENASSISTANT [24]. The LESS dataset comprises 270k data points, from which we randomly select 5% for training. Alpaca is an instruction-following dataset containing 52k data points. OPENWEBTEXT is a recreation of the WEBTEXT[34] corpus, containing approximately 8 million documents.

**Evaluation Dataset.** To evaluate our GREATS, we consider four datasets: MMLU [17], TYDIQA [7], SAMSUM [14], and OPENWEBTEXT [15]. Specifically, MMLU consists of multiple-choice questions covering 57 subjects, including math, computer science, US history, and more. In Table 2, we report accuracy for nine selected subjects: Sociology, US Foreign Policy, Abstract Algebra, Anatomy, Astronomy, Business Ethics, Clinical Knowledge, College Biology, and College Chemistry. TYDIQA is a multilingual question-answering dataset including 11 diverse languages. In our evaluation, we randomly select 500 test data to compute the perplexity and F1 score. The task is to extract the answer to a query from a given passage. SAMSUM is a dialog dataset with the task of summarizing a given dialogue between humans.

**More Training Details.** For the experiment results shown in the main paper, the training hyperparameters are shown below:

1. Finetuning LLAMA-2-7B to MMLU: We finetune LLAMA-2-7B on 5% of the LESS dataset, setting the LoRA rank to 128, LoRA $\alpha$ to 1.0, and dropout to 0.1. The learning rate is set to 2e-5.[3]

2. Finetuning MISTRAL-7B to TYDIQA: We finetune MISTRAL-7B on 40% of the LESS dataset, setting the LoRA rank to 128, LoRA $\alpha$ to 1.0, and dropout to 0.1. The learning rate is set to 1e-5.

3. Finetuning LLAMA-3-8B to SAMSUM: We finetune LLAMA-3-8B on the ALPACA dataset using `Torchtune`[4], setting the LoRA rank to 8, LoRA $\alpha$ to 0.1, and the learning rate to 2e-5.

4. Pretraining GPT-SMALL to OPENWEBTEXT: We pretrain the GPT-SMALL model with a learning rate of 6e-4 and a batch size of 16.[5]

**More Evaluation Details.** To evaluate the accuracy of MMLU and TYDIQA, we follow the LESS paper's approach, using few-shot in-context learning demonstrations. Specifically, we measure the 5-shot accuracy for the MMLU dataset and the 1-shot macro-averaged F1 score for TYDIQA. We also provide examples of evaluation data in Figure 5,6, 7, and 8.

---

**Example of MMLU**

**Question:** The shift from "civil religion" to "common religion" means that:
(A) the increasing bureaucracy of the state has made religion only a marginal part of our lives,
(B) despite the weakening of traditional authority, our everyday lives and 'common sense' remain shaped by religious beliefs and values
(C) religious participation in collective worship may have declined, but people still practise their faiths in private
(D) people are much more likely to discuss their religious beliefs in public, informal settings
**Answer:** B

Figure 5: Example of MMLU

---

[3]Codebase for task 1 and task 2: `https://github.com/princeton-nlp/LESS`
[4]Codebase for task 3: `https://github.com/pytorch/torchtune`
[5]Codebase for task 4: `https://github.com/karpathy/nanoGPT`

Figure 6: Example of TYDIQA

Figure 7: Example of SAMSUM

**Example of OPENWEBTEXT**

**Text:**
A magazine supplement with an image of Adolf Hitler and the title 'The Unreadable Book' is pictured in Berlin. No law bans "Mei...

Figure 8: Example of OPENWEBTEXT

## C Additional Experiment Results

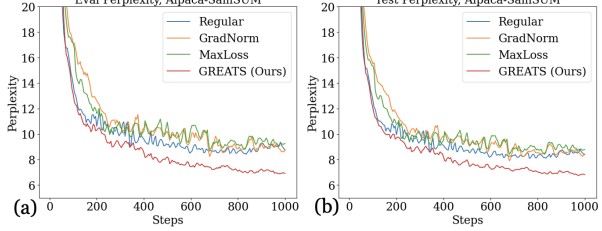

Figure 9: Experimental results on SAMSUM, where we leveraged ALPACA as the training data. Our GREATS method significantly outperforms other approaches.

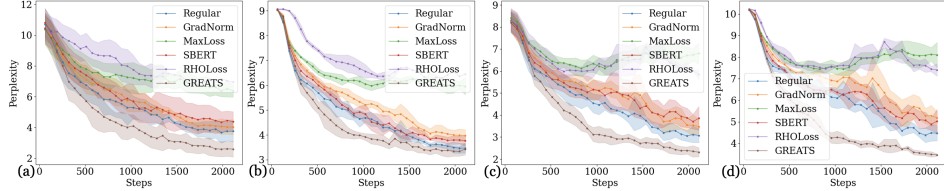

Figure 10: Similar to Figure 2, instead, we use a batch size of **4** and a learning rate of **4e-5**.

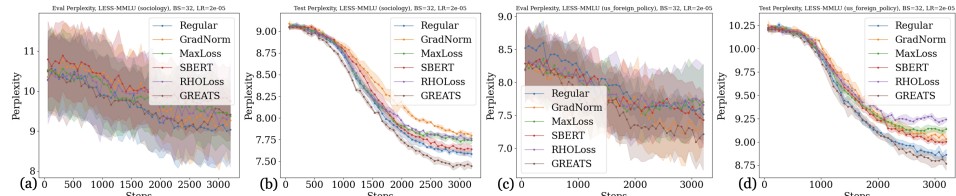

Figure 11: Similar to Figure 2, instead, we use a batch size of **32** and a learning rate of **2e-5**.

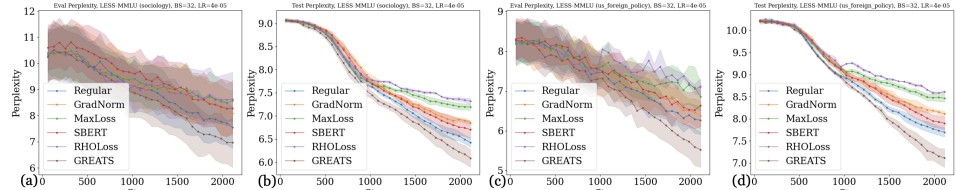

Figure 12: Similar to Figure 2, instead, we use a batch size of **32** and a learning rate of **4e-5**.

## D Broader Impacts

We expect our work to have a positive societal impact. We developed a novel method to facilitate the training process of large language models.

