# OpenReview forum: "GREATS: Online Selection of High-Quality Data for LLM Training in Every Iteration"
_NeurIPS.cc/2024/Conference — NeurIPS 2024 spotlight_

### Official Review · Reviewer_tC7z · 2024-06-27

**Soundness:** 3
**Presentation:** 3
**Contribution:** 2
**Rating:** 6
**Confidence:** 4

**Summary:**

This paper proposes a novel online batch selection algorithm called GREedy Approximation Taylor Selection (GREATS) for training large language models (LLMs). The algorithm aims to improve training convergence speed and generalization performance by selecting informative and diverse examples for model updates. GREATS uses a principled formulation of the online batch selection problem and leverages greedy algorithms and Taylor expansions to approximate the utility of data points. This paper presents extensive experiments on various LLM tasks to demonstrate the effectiveness of GREATS in improving training performance.

**Strengths:**

1) This paper presents a novel online batch selection algorithm that addresses the limitations of existing methods and significantly improves training convergence speed and generalization performance.
2) The algorithm is based on a principled formulation and uses innovative techniques such as greedy algorithms and Taylor expansions to approximate the utility of data points.
3) This paper provides comprehensive evaluations on various LLM tasks, demonstrating the robustness and versatility of GREATS.

**Weaknesses:**

1) The authors only consider MaxLoss and GradNorm as baselines, they do not compare GREATS with other state-of-the-art online batch selection methods.
2) No confidence intervals in any of the tables, which makes it hard to gauge the significance of the accuracy gains.
3) The authors only report accuracy on MMLU and TYDIQA test sets, I cannot find the results of other datasets.

**Questions:**

Why are uniform selection and selection-via-Proxy [*1] not included as baselines?

[*1] Selection via proxy: Efficient data selection for deep learning, ICLR 2020.

**Limitations:**

The authors have addressed the limitations in the main text.

---

> ### Author Rebuttal · Authors · 2024-08-07
>
> We thank the reviewer for the positive feedback!
>
> **Q. [Additional Baseline Comparison.]** *"Why are uniform selection and selection-via-Proxy not included as baselines?"*
>
> **A.** We thank the reviewer for bringing up potentially additional baselines. If we understand correctly, the “uniform selection” baseline corresponds to the “Regular” curve in our current experiments, where the training batches are sampled uniformly at random. For “Selection-via-proxy”, our understanding is that it is a general paradigm for accelerating offline data selection. We'd like to clarify that **offline data selection techniques are not competitors to online batch selection methods** like GREATS. Instead, these approaches are complementary and can be used in conjunction in the training pipeline. Offline methods can provide an initial high-quality dataset, while online methods can further optimize the training process by selecting the most informative batches during runtime. This potential synergy between offline and online data selection is an interesting direction for future research.
>
> As suggested by the reviewers, we have expanded our comparisons to include reference model-based online batch selection techniques [1] and embedding similarity-based batch selection. Our results show that GREATS consistently outperforms these baselines across various hyperparameter settings, further validating its effectiveness. See **Q1 in global response** for a detailed discussion.
>
> [1] Mindermann, Sören, et al. "Prioritized training on points that are learnable, worth learning, and not yet learnt." ICML 2022.
>
> **Q. [Error bar for the experiment results?]**
>
> **A.** We sincerely appreciate the reviewer's insightful suggestion. We have updated the experiment results for key benchmarks with error bars across 3 independent runs.
>
>    |                     | Regular        | GradNorm       | MaxLoss        | GREATS (Ours)  |
>    |---------------------|----------------|----------------|----------------|----------------|
>    | MMLU - sociology    | 62.7% (±0.9)   | 61.0% (±0.5)   | 63.9% (±0.8)   | **66.1% (±0.8)**  |
>    | MMLU - us_foreign_policy | 60.1% (±0.8) | 62.2% (±1.4) | 60.2% (±1.2) | **65.8% (±2.2)**   |
>    | TYDIQA              | 54.3% (±0.3)   | 53.4% (±0.18)  | 54.7% (±0.13)  | **55.0% (±0.3)**   |
>
>
> These updated results with error bars further solidify our findings. GREATS consistently outperforms all baselines across these key benchmarks, even when accounting for run-to-run variability. We've also added error bars to the loss curves for these experiments (see **Q1 in global response**) which show similar results. Due to the computational intensity nature of some experiments, we are in the process of extending this multiple-run analysis to the Alpaca-Samsum and OpenWebText (pretraining) settings. We're committed to providing a complete picture of GREATS' performance across all benchmarks in the revision.
>
> **Q. [Accuracy results for other datasets?]**
>
> **A.** We thank the reviewer for the valuable comment!
>
> **OpenWebText Pretraining:** For our GPT2 pretraining on OpenWebText, we primarily focused on test loss as the most appropriate metric. This choice was made due to the nature of pretraining (where the model is not instruction-tuned) and our computational constraints. Test loss provides a direct measure of the model's ability to predict the next token, which is the core objective of language model pretraining.
>
> **Alpaca-Samsum:** Since Samsum is a summarization task, we evaluated the model's performance using the ROUGE-1 score, which is a standard metric for summarization quality. Here are the additional experiment results on Samsum's test set:
>
> | | Regular  | GradNorm | MaxLoss | GREATS (Ours) |
> |----------|----------|----------|----------|----------|
> | ROUGE-1    | 35.78% (±0.1)  | 35.91% (±0.03) | 35.65%(±0.24)  | **36.74%(±0.46)**    |
>
> As we can see, GREATS outperforms all baselines on this metric, further demonstrating its effectiveness across diverse task types.

---

> > ### Comment · Reviewer_tC7z · 2024-08-14
> >
> > I have read the authors' global and individual responses and acknowledge their point about empirical results, I am happy to increase my rating.

---

### Official Review · Reviewer_SyBn · 2024-07-11

**Soundness:** 3
**Presentation:** 3
**Contribution:** 2
**Rating:** 7
**Confidence:** 4

**Summary:**

The paper introduces a well-motivated online data selection method based on Taylor series expansions (with additional approximations) and apply it to training/fine-tuning LLMs. The paper's results show good performance improvements on fine-tuning tasks, but little gains on pre-training tasks.

**Strengths:**

The paper is very well motivated, and tackles a problem which has clear significance to large scale model training.

The problem setup is very intuitive and straightforward: select the data points now which would maximize utility after one model update step.

The paper is also relatively well written. The method is clearly derived and explained -- most/each introduced approximation is accompanied by a clear interpretation.  I most enjoyed reading section 3.2 -- splitting out the importance score of a sample and correcting it based on the scores of previously seen samples is very neat.

The paper includes several relevant experiments (most on fine-tuning) -- using a few relevant baselines. The results appear strong, in general, when compared to the selected baselines.

**Weaknesses:**

I gave the paper a rating of 5 mostly because of the limited evaluation. Simple baselines are just missing from the paper, e.g.: (1) a classifier to detect similarity to validation data and (2) rho loss. I would be happy to increase my rating if these baselines are added to the paper.

To clarify, regarding rho loss, I do not find the argument in line 262-263 at all convincing. In a separate pass, you can just label the training data with per token losses from a pre-trained reference model. No extra training flops needed over the "Max Loss" baseline.

Note that influence functions [Koh & Liang] (in the context of LLMs too [Grosse et al.]) are not discussed anywhere, even though they are closely related with the method proposed here. The difference is that influence functions look at final performance and the paper here looks at 1-step performance. It would be great to see what happens when consider n-step unrolls, i.e., and use the influence functions formulation -- of course, this could be left specifically for future work.

The runtime complexity (i.e. number of flops) of the GREATS algorithm is missing (under every condition) and is not compared to that of regular training. Indeed runtime comparisons (i.e. in seconds) are useful, they are a bit incomplete.

A minor point is that the argument in lines 28-29 is tenuous at best ("Moreover, online batch selection operates on smaller batches of data, reducing the need for cumbersome data preprocessing and enabling more efficient use of computational resources compared to static data selection methods that process the entire dataset upfront".) It is much more convenient to pre-cache the data when training these models. At training time, one really just wants to use accelerators as efficiently as possible.

[Koh & Liang] Understanding Black-box Predictions via Influence Functions
[Grosse] Studying Large Language Model Generalization with Influence Functions

**Questions:**

- It is unclear when the Hessian is set to the identity in all GREATS experiments. I assume H=I always, as otherwise the runtime would blow up. Is this assumption correct?

- How did you choose the (up to 16) validation dataset points? Randomly?

**Limitations:**

The authors have moderately addressed the limitations of their work. I've pointed out what else I was expecting to see in the weaknesses section.

---

> ### Author Rebuttal · Authors · 2024-08-07
>
> We thank the reviewer for the positive comments!
>
> **Q. [Additional Baseline Comparison.]**
>
> **A.** We sincerely appreciate the reviewer's feedback and suggestions for additional baselines. We have taken these recommendations seriously and have expanded our comparisons accordingly. **(1) Reference model-based online batch selection (RHO Loss) [1]**: We have implemented and evaluated the RHO Loss method as suggested. **(2) Similarity to validation data:** Regarding the suggestion of *"a classifier to detect similarity to validation data,"* we note that our validation sets are very small, which may not be sufficient to train a robust classifier. Instead, we implemented a method that selects the training batch based on embedding similarity to the validation data, using Sentence-BERT embeddings. This approach serves as a proxy for the suggested classifier-based method while being applicable to our small validation set scenario.
>
> Our expanded experiments show that GREATS consistently outperforms these additional baselines across various hyperparameter settings, further validating its effectiveness. We have included a detailed discussion of these results in **Q1 in global response**.
>
> [1] Mindermann, Sören, et al. "Prioritized training on points that are learnable, worth learning, and not yet learnt." ICML 2022.
>
> **Q. [Relationship to the influence function.]**
>
> **A.** We appreciate the reviewer's insightful comment regarding the relationship between our proposed method and influence functions. Influence functions typically assume strong convexity of the loss landscape and convergence to a local optimum. In contrast, our method makes no such assumptions, making it more suitable for the highly non-convex landscapes encountered in large language model training. While influence functions focus on the impact of training examples on the final model performance, our method explicitly considers the immediate impact on the next training step. This allows us to capture the dynamic nature of the training process and adapt to the model's current state. We will incorporate this discussion into the related work section.
>
> **Q. [Runtime complexity of GREATS?]**
>
> **A.** We appreciate the valuable suggestion! Here’s the complexity analysis for GREATS: Assuming that the batch size is $B$ and the number of model parameters is $p$, the complexity of backpropagation is $O(Bp)$.
> The additional cost of the "ghost" dot-product involves computing the pairwise dot-product between each data point’s activation (or output derivative), which are of dimension $O(\sqrt{p})$, and then taking the scalar product between the inner product results on activations and output derivatives. The step of computing the pairwise activation (or output derivative) dot-products has a runtime complexity of $O(B^2 \sqrt{p})$, resulting in a $B \times B$ matrix. Taking the element-wise product between the two $B \times B$ matrices has a runtime complexity of $O(B^2)$. Hence, the overall runtime complexity is $O(Bp) + O(B^2 \sqrt{p})$. In practice, since the batch size $B$ is significantly smaller than the number of parameters $p$, the additional runtime $O(B^2 \sqrt{p})$ is negligible compared to the backpropagation complexity $O(Bp)$. This analysis explains why our empirical runtime measurements (in seconds) show that GREATS performs comparably to regular training. We will incorporate this detailed complexity analysis into our paper to provide a more comprehensive understanding of the algorithm's efficiency. Thank you for highlighting this important aspect!
>
>
>
> **Q.** “*A minor point is that the argument in lines 28-29 is tenuous at best ("Moreover, online batch selection operates on smaller batches of data, reducing the need for cumbersome data preprocessing and enabling more efficient use of computational resources compared to static data selection methods that process the entire dataset upfront".) It is much more convenient to pre-cache the data when training these models. At training time, one really just wants to use accelerators as efficiently as possible.*”
>
> **A.** We appreciate the reviewer’s valuable comments. We agree with the reviewer and acknowledge that offline data selection provides the convenience for not modifying training pipeline. We will include this point and modify the Introduction accordingly. On the other hand, online batch selection offers unique advantages in terms of adaptability to the model's learning progress. This can be particularly valuable in scenarios where the relevance of data may change as training progresses. Furthermore, offline and online selection methods are not mutually exclusive and can be complementary. A hybrid approach, leveraging both pre-cached, pre-selected data and dynamic online selection, could potentially offer the best of both worlds.
>
> **Q. [Hessian approximation used in the experiments?]**
>
> **A.** In all of our experiments, we use identity matrix as the approximation to the Hessian matrix, and we found the performance is already good. Future work can further explore the use of more advanced techniques for Hessian approximation, such as K-FAC [1].
>
> [1] Martens, James, and Roger Grosse. "Optimizing neural networks with kronecker-factored approximate curvature." ICML 2015.
>
> **Q. [How to choose the validation data?]**
>
> **A.** In the standard benchmarks such as MMLU, the validation data is being provided for each subject of size 5. For the benchmarks where the validation set size is large, we sample the validation set used in the experiments uniformly at random. We will incorporate this additional detail into our experiment setting section.

---

### Official Review · Reviewer_oEUR · 2024-07-13

**Soundness:** 3
**Presentation:** 3
**Contribution:** 3
**Rating:** 7
**Confidence:** 3

**Summary:**

The paper presents an online adaptive subset selection framework GREATS which acts as a principled and efficient online batch selection. The authors further showcase their proposal's utility towards training for Large language models training thereby better performance during training

**Strengths:**

- The paper for is well written, easy to follow.
- Contributions are well formulated and clearly stated.
- The paper's proposal is well-suited towards better efficient training in Large language model training as authors show extensive experiments in practical utility of their proposal towards efficient LLM training.

**Weaknesses:**

Notation could be a bit better:
- For instance under definition of $\textbf{Utility Function}$, $U^{(t)}: \mathrm{R}^d \rightarrow \mathrm{R}^d$ denotes the utility function mapping the training set to an optimal batch.
- vector notations should be bold

Many adaptive subset selection techniques have been shown to be either submodular or weakly submodular. As in either case, submodular functions provide better functionality. It would have been interesting to see if the proposed utility function which is essentially the loss w.r.t validation data satisfies some form of submodularity. Some form of theoretical justification would be highly appreciated.


Experiment comparison wise:
- Regarding baselines, I am wondering why authors have not tried any baseline comparison against submodular subset selection techniques. Some examples: https://proceedings.mlr.press/v139/killamsetty21a. Subset selection techniques is often to show better in many cases.

I am willing to increase my rating if this questions can be addressed.

**Questions:**

NA

**Limitations:**

The authors have clearly mentioned about their proposed limitations.

---

> ### Author Rebuttal · Authors · 2024-08-07
>
> **Q. [Notation]**
>
> **A.** We thank the reviewer for the suggestions on the notations, which we will incorporate into our revision. We note that the utility function $U^{(t)}: R^d \rightarrow R$ maps to the model performance change in $t$th step instead of the optimal batch.
>
> **Q. [Is the proposed utility function close to submodular?]**
>
> **A.**
> We sincerely appreciate the reviewer's insightful question. We conducted additional experiments and in **Global response’s Figure 5 (c)**, we plot how the loss change in a single gradient update step, i.e., our utility function $U^{(t)}(S)$, against  varying input batch sizes $|S|$. We observed that as batch sizes increase, the average loss change (computed across randomly sampled subsets of fixed sizes) indeed exhibits a "diminishing returns" trend. This behavior is consistent with the general properties of submodular functions. However, we also noted significant variance in the utilities, especially for smaller batch sizes. This high variance suggests that strict submodularity may not hold. The observed "diminishing returns" trend aligns with the intuition behind submodular functions and provides some justification for the effectiveness of our greedy selection approach. While our utility function may not be strictly submodular, we conjecture that it could potentially fall into the category of weakly submodular functions.
>
> We will incorporate this analysis into our paper, which might open up interesting theoretical directions for future work.
>
> **Q. [Discussion on submodular subset selection techniques]**
>
> **A.** We sincerely appreciate the reviewer for bringing our attention to the relevant literature on submodular subset selection techniques.
> To the best of our knowledge, gradient coreset-based online data selection algorithms, including those utilizing submodular optimization (like [1]), generally face scalability challenges when applied to the setting of foundation models. The primary limitation stems from their requirement to compute (or even store) per-sample gradient vectors, which becomes prohibitively expensive in terms of both computation (or memory) for large-scale models with millions or billions of parameters. For instance, if we understand correctly, to optimize the resulting error in [1], the computation of $Err$ requires computing per-sample gradient vectors. An interesting future avenue is to explore whether our "ghost inner product" technique can be further extended to address the computational problems in this line of work.
> We will incorporate these references and discussion into our related work section in the revision.
>
> **Additional baselines:** As suggested by other reviewers, we have expanded our comparisons to include reference model-based online batch selection techniques [2] and embedding similarity-based batch selection. Our results show that GREATS consistently outperforms these baselines across various hyperparameter settings, further validating its effectiveness. See **Q1 in global response** for a detailed discussion.
>
> [1] Killamsetty, Krishnateja, et al. "Grad-match: Gradient matching based data subset selection for efficient deep model training." ICML 2021
>
> [2] Mindermann, Sören, et al. "Prioritized training on points that are learnable, worth learning, and not yet learnt." ICML 2022.

---

> > ### Comment · Reviewer_oEUR · 2024-08-14
> > **Response by Reviewer**
> >
> > I thank the authors for providing clarification to the questions I had. Considering the responses and doing another detailed pass at the paper, I am happy to increase my rating,

---

### Official Review · Reviewer_vNzN · 2024-07-13

**Soundness:** 3
**Presentation:** 2
**Contribution:** 3
**Rating:** 7
**Confidence:** 3

**Summary:**

The paper proposes a new online batch selection algorithm, GREATS, which frames batch selection as an optimization problem: specifically, selecting a subset of the training data that maximally reduce the validation loss when used as part of training. It uses a Taylor approximation of the effects of training on a given data point to estimate the value of each data point for the purposes of training, including a second order Hessian term that partially accounts for potential interactions between the data points of the batch affecting their collective value as training data, then greedily selects a batch of training data to minimize validation loss. Further, the paper propose the 'ghost inner product' method of efficiently computing pairwise gradient inner products with only a single step of backpropegation.

The paper tests their method in a variety of finetuning settings on Llama-2-7B, Mistral-7B, and Llama-3-8B (trained on either LESS or Alpaca datasets and tested on MMLU, TydiQA, or SamSUM), as well as for pretraining a GPT2 base model from scratch on OpenWebText data. It uses some fairly simple baselines (the "regular" unbiased sampling of batches, sampling high loss data points, and sampling high gradient norm data points), and finds that GREATS consistently outperforms them. Finally, the authors address the questions of sensitivity to validation data set size and of runtime, finding that GREATS is pretty robust to using smaller validation sets and that their implementation of GREATS has little computational overhead.

**Strengths:**

**Originality**:
The clear formulation of the objective for online batch selection, the Taylor approximation-based approach to efficiently optimize that objective, and the ghost inner product method all seem like potentially original and valuable contributions.

**Quality**:
The paper largely seems well put together and executed, though with some potential issues related to a lack of strong baselines.

**Clarity**:
The paper was generally clear, with some grammatical issues or weird phrasing. E.g.,
Line 13: "the extensive training times" --> "their extensive training times"
Line 59: "updating model" --> "updating the model"
Line 62: "one-step gradient" --> "one step of gradient" or "single-step gradient"
Line 87: ", and update" --> "and updates"
Line 88: "where" --> ", where"
and various other places.

However, the introduction of "utility functions" seems superfluous. It seems like the authors can just talk about the optimization objectives of various batch selection methods directly. E.g., "We search for training datapoints that minimize the validation loss" vs "We propose to set the utility function to be the validation loss, and then search for training data points that optimize this utility function".

**Significance**:
The significance of this contribution is hard to judge due to 1) potentially weak baselines, and 2) lack of engagement with more contemporary prior work.
However, it seems potentially high, due to the importance of the domain, and because the approach seems simple and intuitively well-motivated.

**Weaknesses:**

I think the biggest weakness of this paper is the lack of stronger baselines to compare against. E.g., both of the following seem like stronger contenders than the max loss / max grad norm approaches:
- https://proceedings.neurips.cc/paper_files/paper/2023/hash/dcba6be91359358c2355cd920da3fcbd-Abstract-Conference.html
- https://proceedings.neurips.cc/paper_files/paper/2023/hash/6b9aa8f418bde2840d5f4ab7a02f663b-Abstract-Conference.html

If I'm considering using some online batch selection method from the literature, max loss / max grad norm don't really seem like they'd even be in the running. So comparing against them doesn't really help establish to be that GREATS would be the most appropriate choice.

**Questions:**

Do the authors use SGD or Adam in their experiments? The limitations section mentions the ghost inner product technique doesn't work for Adam, but says "using SGD as a proxy for Adam has proved to be effective in our experiment." Does this mean the authors compute the ghost inner product for SGD, but then just use the results for Adam? (The learning rates seem to suggest so).

I have concerns about the optimality of the training setups. The paper does not appear to describe any sort of hyper parameter tuning for the finetuning or pretraining. How were these values selected? Additionally, given that SGD (or Adam?) with small batch sizes is a nonstandard choice for such experiments, how would the authors argue that their results can be expected to be relevant to a setting that better matches current SOTA (e.g., with Adam and large batch sizes)? Can we expect the method's benefits to hold for larger batch training, given that batch selection should become less impactful as batch size increases (e.g., going to zero once we reach full-batch training).

From what I understand, the authors always select batches using validation data drawn from the same distribution as the testing data. In realistic training settings, one usually does not know the exact distribution form which the downstream test data will be drawn. Thus, I wonder if the authors could address the question of what happens when the validation data used during online batch selection is partially "off distribution" for the actual test data?

**Limitations:**

Lacking an implementation of the ghost inner product method for Adam and similar state of the art optimizers may be a significant limitation of this method? It depends on whether the authors were forced to use SGD in their experiments.

---

> ### Author Rebuttal · Authors · 2024-08-07
>
> We thank the reviewer for the positive comments!
>
> **Q. [Relationship between online batch selection and offline data selection / data mixture optimization techniques [1, 2]? & Additional baseline comparison]**
>
> **A.**
> We appreciate the reviewer highlighting the relevant literature on offline data selection [1] and data mixture optimization [2]. We'd like to clarify that **offline data selection techniques are not direct competitors to online batch selection methods** like GREATS. Instead, these approaches are complementary and can be used in conjunction in the training pipeline. Offline methods can provide an initial high-quality dataset, while online methods can further optimize the training process by selecting the most informative batches during runtime. This potential synergy between offline and online data selection is an interesting direction for future research.
>
> **Additional baselines:** As suggested by other reviewers, we have expanded our comparisons to include reference model-based online batch selection techniques [3] and embedding similarity-based batch selection. Our results show that GREATS **consistently outperforms** these baselines across various hyperparameter settings, further validating its effectiveness. See **Q1 in global response** for a detailed discussion.
>
> [1] Xie, Sang Michael, et al. "Data selection for language models via importance resampling." NeurIPS 2023.
>
> [2] Xie, Sang Michael, et al. "Doremi: Optimizing data mixtures speeds up language model pretraining." NeurIPS 2024.
>
> [3] Mindermann, Sören, et al. "Prioritized training on points that are learnable, worth learning, and not yet learnt." ICML 2022.
>
> **Q. [Experiment with different hyperparameters]**
>
> **A.** As suggested, we have performed experiments with different choices of learning rates and batch sizes. Our results show that GREATS **consistently performs** well across all the hyperparameter configurations we considered. This robustness suggests that GREATS can maintain its effectiveness without requiring extensive hyperparameter tuning. An ablation study on the number of validation points in Figure 3 (a)-(b) of the paper. This study demonstrates that GREATS can perform effectively even with a small number of validation points (2 to 5), which is a crucial aspect of its practical applicability.
>
> **Q. [Potential distribution shift between validation and test distribution?]**
>
> **A.**
> We appreciate the reviewer's insightful question. **Subsampling validation data** is one solution here. This subsampling approach effectively reduces overfitting to the validation set and introduces some variability in the selection process, which can help mitigate the impact of potential distribution mismatches.
>
> We acknowledge that the availability of a high-quality validation set is a limitation of GREATS. However, most state-of-the-art data selection (e.g., [1, 2]) and online batch selection techniques (e.g., [3]) rely on a high-quality validation set. This is a common approach in the field to guide the selection process toward data that aligns with the target domain or task. In finetuning tasks, practitioners typically have a small set of high-quality, task-specific data that can serve as a validation set. Our experiments demonstrate that **GREATS can perform well even with a very small validation set** (as few as 2-5 samples), which makes the requirement more practical to fulfill in many scenarios. We stress that while GREATS does require a validation set, it avoids other limitations of existing methods, such as the need for reference models, making it more practical for large-scale training scenarios.
>
> [1] Xia, Mengzhou, et al. "LESS: Selecting Influential Data for Targeted Instruction Tuning." ICML 2024.
>
> [2] Xie, Sang Michael, et al. "Data selection for language models via importance resampling." NeurIPS 2023.
>
> [3] Mindermann, Sören, et al. "Prioritized training on points that are learnable, worth learning, and not yet learnt." ICML 2022.
>
> **Q [Do the experiments use Adam optimizer?]**
>
> **A.** **In all of our experiments, we use AdamW as the optimizer for model training.** This aligns with common practices in training large language models. We acknowledge that our theory for GREATS is derived in terms of SGD. While the theory can be extended to Adam, the efficient "ghost" inner-product technique we developed is specifically tailored for SGD and is challenging to directly adapt to Adam due to its more complex update rule and maintained moments. Hence, we adopt a hybrid approach where we select the valuable batch based on the theory derived from SGD, but perform the actual training updates using AdamW. This approach allows us to benefit from the computational efficiency of our SGD-based selection method while still leveraging the optimization benefits of AdamW for training. Despite this theoretical mismatch, our experiments demonstrate that this hybrid approach performs well in practice. The consistent improvements over baselines across various models and tasks suggest that the SGD-based utility function provides a good proxy for data importance, even when Adam is used for training. We recognize the potential for further improvement by aligning the batch selection more closely with Adam-style updates.

---

> > ### Comment · Reviewer_vNzN · 2024-08-13
> > **Response to author rebuttal**
> >
> > I thank the authors for their detailed responses to the points raised.
> >
> > The authors' additional experiments with their new baselines have addressed the primary limitation I saw with this work, and I have accordingly raised my soundness score from 2 to 3 and my overall rating from 6 to 7.
> >
> > Although it is good to see that the proposed method is effective across a number of different hyper parameter settings, this does not address my question regarding the effectiveness of the method in the high batch size setting. From the figures provided, it looks like GREATS with batch size may be 4 a relatively larger improvement over the baselines than GREATS with a batch size of 32, and the rest of the baseline methods also become more similar to each other in the batch size 32 experiments (though this could also just be because the batch size 32 runs have far worse perplexity for some reason?). If GREATS does become less of a relative improvement as batch size increases, I strongly encourage the authors to explicitly discuss this fact in the paper (while of course highlighting that this should be an issue for any online batch selection algorithm).
> >
> > The authors also suggested that they might subsample validation data to improve generalization to unknown downstream tasks. I agree it might, but there aren't really empirical results to support such a possibility. On the other hand, I acknowledge that ensuring generalization to unknown domains is challenging, so perhaps this is the best that can reasonably be done in such a timeframe.

---

### Official Review · Reviewer_2jYA · 2024-07-13

**Soundness:** 3
**Presentation:** 3
**Contribution:** 3
**Rating:** 5
**Confidence:** 4

**Summary:**

This paper proposes GREATS (GREedy Approximation Taylor Selection), a novel online batch selection algorithm for large language model (LLM) training. GREATS aims to improve training efficiency by dynamically selecting the most informative data points from each training batch, based on their potential to reduce validation loss. The authors formulate the batch selection problem as set function optimization and leverage Taylor expansions to efficiently approximate the impact of each data point on validation loss. To address the computational overhead of calculating per-sample gradients, they introduce a technique called "ghost inner-product." Experimental results demonstrate that GREATS consistently accelerates training convergence and improves generalization performance across various language modeling tasks.

**Strengths:**

*  GREATS is grounded in a principled set function optimization framework, aiming to directly optimize model performance on a held-out validation set. This contrasts with heuristic-based methods, providing a stronger theoretical foundation.
*  Leveraging Taylor expansions to approximate the marginal gain of data points is a clever approach that significantly reduces computational complexity. This makes GREATS more practical for large-scale LLM training compared to methods requiring repeated model updates and validation evaluations.
* The proposed "ghost inner-product" technique for efficient gradient inner-product calculation is innovative and addresses a major computational bottleneck in online batch selection. This technique has the potential for broader applicability in machine learning beyond data selection.

**Weaknesses:**

* As also pointed out by the authors, GREATS relies on a small set of clean validation data, which may not always be readily available in practical scenarios, especially for pretraining. Classifying a point as noisy and not of interest just because a similar point doesnt exist in validation might not be ideal, and I am not sure how GREATS could be handling this.

* Approximation Accuracy: While the use of Taylor expansions for approximation is efficient, the accuracy of these approximations can be affected by factors like learning rate and non-linearity of the model. Further analysis and discussion on the potential limitations of these approximations would strengthen the paper.

* Limited Comparison: The paper only compares GREATS against simple baseline methods (Regular, MaxLoss, GradNorm) and excludes comparison with reference-model-based methods. While these are computationally expensive, including such comparisons might be beneficial to the larger research community.

*  The performance of GREATS is likely sensitive to various hyperparameters like the learning rate, batch size, and the number of validation data points. The paper lacks a systematic study of these hyperparameter sensitivities.

* As pointed out by the authors, the "ghost-inner product" technique works best in tandem with SGD, but not with Adam or other optimizers. Thus, it is unclear if these models used Adam while training, but just SGD for the utility function computation, or if the entire model was trained through SGD itself. I believe, Adam with weight decay has become a common optimizer for these LLMs and the findings if presented by training through SGD might not translate to the practical setting where Adam or AdamW is used. I believe further clarification might be required here.

**Questions:**

Please look at weaknesses.

---

> ### Author Rebuttal · Authors · 2024-08-07
>
> We thank the reviewer for the nice words!
>
> **Q [Requirement of validation set]**
>
> **A.** We acknowledge that the availability of a high-quality validation set is a limitation of GREATS. However, most state-of-the-art data selection (e.g., [1, 2]) and online batch selection techniques (e.g., [3]) rely on a high-quality validation set. This is a common approach in the field to guide the selection process toward data that aligns with the target domain or task. In finetuning tasks, practitioners typically have a small set of high-quality, task-specific data that can serve as a validation set. **Validation data for pretraining:** We agree that for pretraining scenarios, obtaining a representative validation set can be more challenging. However, even in these cases, it's often possible to curate a small set of diverse, high-quality samples that help for data selection [4, 5, 6]. Our experiments demonstrate that **GREATS can perform well even with a very small validation set** (as few as 2-5 samples), which makes the requirement more practical to fulfill in many scenarios. Furthermore, GREATS selects points based on their potential to improve performance on the validation set instead of simply filtering data points that are similar to the validation data. It's worth noting that while GREATS does require a validation set, it avoids other limitations of existing methods, such as the need for reference models, making it more practical for large-scale training scenarios.
>
> [1] Xia, Mengzhou, et al. "LESS: Selecting Influential Data for Targeted Instruction Tuning." ICML 2024
>
> [2] Xie, Sang Michael, et al. "Data selection for language models via importance resampling." NeurIPS 2023
>
> [3] Mindermann, Sören, et al. "Prioritized training on points that are learnable, worth learning, and not yet learnt." ICML 2022
>
> [4] Tirumala, Kushal, et al. "D4: Improving llm pretraining via document de-duplication and diversification." NeurIPS 2024
>
> [5] Maini, Pratyush, et al. "Rephrasing the Web: A Recipe for Compute and Data-Efficient Language Modeling."
>
> [6] FineWeb: decanting the web for the finest text data at scale.
>
> **Q [Approximation accuracy of Taylor expansion]** *“While the use of Taylor expansions for approximation is efficient, the accuracy of these approximations can be affected by factors like learning rate and non-linearity of the model. Further analysis and discussion on the potential limitations of these approximations would strengthen the paper.”*
>
> **A.** We thank the reviewer for the valuable suggestions. We conducted additional experiments and examined two key correlations in **Figure 5 in Global response**: **(a)** The correlation between the actual validation loss change in a single gradient update step and the loss change predicted by the sum of training gradients taking dot product with the validation point (i.e., the first term in Equation (4)). **(b)** The correlation between the actual validation loss change in a single gradient update step and the loss change predicted by both the gradient dot product sum and the Hessian interaction term, as shown in Equation (4). As we can see, if we do not incorporate the interaction term, the Pearson correlation coefficient is approximately 0.75. Figure (b) shows that the correlation coefficient improved by including the Hessian interaction term to approximately 0.84. These results demonstrate that our approximations capture a significant portion of the actual loss change, with the inclusion of the Hessian interaction term providing a notable improvement in accuracy.
>
> **Q [Comparison with reference-model-based methods]**
>
> **A.** We have conducted additional experiments using the RHO-Loss method from [1] as a representative baseline for reference-model-based approaches. Due to time and computational constraints, we used Llama-3.1-8B-Instruct as the reference model (with Llama2-7B as the target model), prioritizing training on data points with high RHO loss ($L(x; f_{tgt}) - L(x; f_{ref})$ where $f_{tgt}$ is the target model and $f_{ref}$ is the reference model). Surprisingly, the RHO-Loss method performed poorly, showing the worst performance across almost all settings. We hypothesize two main reasons here: **(1) Noise and scale discrepancy:** The loss value from the current target model $L(x; f_{tgt})$ is highly noisy and may be of a different scale compared to the reference model. This discrepancy could result in $L(x; f_{ref})$ having minimal impact on the batch selection results. **(2) Reference model limitations:** Due to computational and time constraints, we used a relatively small pretrained model as the reference. This may not adequately reflect the "learnability" or "difficulty" of the training examples, which is crucial for effective data selection. These findings highlight a key advantage of GREATS: it offers a more efficient and flexible alternative that can perform well without relying on reference models, which can be computationally expensive or may not be available in practice.
>
> [1] Mindermann, Sören, et al. "Prioritized training on points that are learnable, worth learning, and not yet learnt." ICML 2022.
>
> **Q [Experiment with additional baselines & different hyperparameters]**
>
> **A.** As suggested, we have performed experiments with additional baselines and different choices of learning rates / batch sizes. The results are in **Global response Q1**. The results show that GREATS **consistently performs well** across all the hyperparameter configurations we considered. This robustness suggests that GREATS can maintain its effectiveness without requiring extensive hyperparameter tuning. An ablation study on the number of validation points in Figure 3 (a)-(b) of the paper. This study demonstrates that GREATS can perform effectively even with a small number of validation points (2 to 5), which is a crucial aspect of its practical applicability.

---

### Author Rebuttal · Authors · 2024-08-07

We thank all the reviewers for the positive assessments!

**Q1 [Additional baseline comparisons & hyperparameter choices]**

**A.** We appreciate the reviewers' suggestions for additional baseline comparisons and hyperparameter sensitivity analysis. In response, we have conducted extensive additional experiments in **Figure 1-4 in the attached pdf**.

**Additional Baseline Comparisons.**
- **Reference model-based online batch selection [1] (Reviewers 2jYA, SyBn)**: We implemented the RHO-Loss method from [1] as a representative baseline for reference model-based approaches. As suggested by Reviewer SyBn, we use a pretrained LLM as the reference model. Due to time and computational constraints, we used Llama-3.1-8B-Instruct as the reference model (with Llama2-7B as the target model), prioritizing training on data points with high RHO loss ($L(x; f_{tgt}) - L(x; f_{ref})$ where $f_{tgt}$ is the target model and $f_{ref}$ is the reference model). Surprisingly, the RHO-Loss method performed poorly, showing the worst performance across almost all settings. We hypothesize two potential reasons here: **(1) Noise and scale discrepancy:** The loss value from the current target model $L(x; f_{tgt})$ is highly noisy and may be of a different scale compared to the reference model. This discrepancy could result in $L(x; f_{ref})$ having minimal impact on the batch selection results. **(2) Reference model limitations:** Due to computational and time constraints, we used a relatively small pretrained model as the reference. This may not adequately reflect the "learnability" or "difficulty" of the training examples, which is crucial for effective data selection. These findings highlight a key advantage of GREATS: it offers a more efficient and flexible alternative that can perform well **without relying on reference models**, which may not be available in practice.
- **Similarity-based batch selection (Reviewer SyBn)**: We implemented a baseline that selects training batches based on their similarity to validation data using Sentence-BERT embeddings. This approach also did not outperform GREATS.

**Hyperparameter Sensitivity Analysis.** We repeated MMLU experiments with various hyperparameter settings, including different batch sizes and learning rates. Each experiment was run with 3 different random seeds, and we have included error bars in our plots to reflect the variability. Across these expanded comparisons and varied hyperparameter settings, GREATS **consistently outperformed** all baselines. This robust performance across different scenarios further validates the effectiveness and stability of our proposed method.

[1] Mindermann, Sören, et al. "Prioritized training on points that are learnable, worth learning, and not yet learnt." ICML 2022.

**Q2 [Do the experiments use Adam optimizer?] (for Reviewer 2jYA and vNzN)**

**A.** **In all of our experiments, we use AdamW as the optimizer for model training.** This aligns with common practices in training large language models. We acknowledge that our theory for GREATS is derived in terms of SGD. While the theory can be extended to Adam, the efficient "ghost" inner-product technique we developed is specifically tailored for SGD and is challenging to directly adapt to Adam due to its more complex update rule and maintained moments. Hence, we adopt a hybrid approach where we select the valuable batch based on the theory derived from SGD, but perform the actual training updates using AdamW. This approach allows us to benefit from the computational efficiency of our SGD-based selection method while still leveraging the optimization benefits of AdamW for training. Despite this theoretical mismatch, our experiments demonstrate that this hybrid approach performs well in practice. The consistent improvements over baselines across various models and tasks suggest that the SGD-based utility function provides a good proxy for data importance, even when Adam is used for training. We recognize the potential for further improvement by aligning the batch selection more closely with Adam-style updates.

---

### Decision · Program_Chairs · 2024-09-25

**Decision:**

Accept (spotlight)

**Comment:**

This paper tackles a significant problem: dynamically selecting data batches during training to improve efficiency for large-scale LLM training. The proposed algorithm, GREATS, formulates batch selection as an optimization problem and employs greedy algorithms and Taylor expansions to approximately t solve the optimization problem. Additionally, it proposes a "ghost inner product" method for efficiently computing pairwise gradient inner products within a single backprop step. The empirical results are comprehensive and demonstrate strong performance in improves training convergence speed and generalization performance when compared to relevant baselines.